# RNA-controlled nucleocytoplasmic shuttling of mRNA decay factors regulates mRNA synthesis and a novel mRNA decay pathway

Shiladitya Chattopadhyay[1], Jose Garcia-Martinez[2], Gal Haimovich [1,7], Jonathan Fischer[3], Aya Khwaja[1], Oren Barkai[1], Silvia Gabriela Chuartzman[4], Maya Schuldiner [4], Ron Elran[1], Miriam I. Rosenberg [1,8], Shira Urim[1], Shubham Deshmukh[1], Katherine E. Bohnsack [5,6], Markus T. Bohnsack [5,6], Jose E. Perez-Ortin [2] & Mordechai Choder [1] ✉

mRNA level is controlled by factors that mediate both mRNA synthesis and decay, including the 5' to 3' exonuclease Xrn1. Here we show that nucleocytoplasmic shuttling of several yeast mRNA decay factors plays a key role in determining both mRNA synthesis and decay. Shuttling is regulated by RNA-controlled binding of the karyopherin Kap120 to two nuclear localization sequences (NLSs) in Xrn1, location of one of which is conserved from yeast to human. The decaying RNA binds and masks NLS1, establishing a link between mRNA decay and Xrn1 shuttling. Preventing Xrn1 import, either by deleting *KAP120* or mutating the two Xrn1 NLSs, compromises transcription and, unexpectedly, also cytoplasmic decay, uncovering a cytoplasmic decay pathway that initiates in the nucleus. Most mRNAs are degraded by both pathways - the ratio between them represents a full spectrum. Importantly, Xrn1 shuttling is required for proper responses to environmental changes, e.g., fluctuating temperatures, involving proper changes in mRNA abundance and in cell proliferation rate.

A high degree of regulation of mRNA levels is a critical feature of gene expression in any living organism. In recent years, we and other investigators have discovered reciprocal adjustments between the overall rates of mRNA synthesis and degradation, named "mRNA buffering", which maintain proper concentrations of mRNAs[1-7]. We have previously demonstrated that in the budding yeast *Saccharomyces cerevisiae* (from herein termed "yeast"), a number of factors, known to regulate or execute mRNA degradation in the cytoplasm, e.g., Xrn1 (alias – Kem1), Dcp2, Pat1, Lsm1[8,9], shuttle between the nucleus and the cytoplasm, by an unknown mechanism[3]. The mRNA buffering

mechanism is not restricted to factors recognized as mRNA decay factors (DFs). It also includes components of the transcription apparatus. We demonstrated that Pol II regulates mRNA translation and decay by mediating Rpb4/7 co-transcriptional binding to Pol II transcripts[10-12], a process we named "mRNA imprinting"[13,14]. More "classical" yeast DFs – components of the Ccr4-Not complex - also imprint mRNA and regulate mRNA export, translation and decay[15,16]. Even a promoter-specific transcription factor, Rap1, can control mRNA decay via mRNA imprinting[17]. Thus, we hypothesize that mechanisms that regulate the cellular localization of factors that mediate mRNA

[1]Department of Molecular Microbiology, Rappaport Faculty of Medicine, Technion-Israel Institute of Technology, Haifa 31096, Israel. [2]Instituto de Biotecnología y Biomedicina (Biotecmed), Universitat de València; Burjassot, Valencia 46100, Spain. [3]Department of Biostatistics, University of Florida, Gainesville, FL 32611, USA. [4]Department of Molecular Genetics, Weizmann Institute of Science, Rehovot 7610001, Israel. [5]Institute for Molecular Biology, University Medical Center Göttingen Georg-August-University, Göttingen, Germany. [6]Göttingen Centre for Molecular Biosciences, Georg-August-University, Göttingen, Germany. [7]Present address: Department of Molecular Genetics, Weizmann Institute of Science, Rehovot 7610001, Israel. [8]Present address: Department of Genetics, Hebrew University of Jerusalem, Givat Ram, Jerusalem 9190401, Israel. ✉e-mail: choder@technion.ac.il

buffering are critical for the linkage between their two opposing activities, in either synthesis or degradation.

Here we show that Xrn1 nuclear import is mediated by two NLSs that are recognized by the importin Kap120. Xrn1 import is controlled by the decaying RNA, establishing a link between the cytoplasmic mRNA decay and the nuclear mRNA synthesis. We name the affected mRNAs as "Kem1 import-sensitive (Kis) mRNAs" and their decay pathway as "Kis pathway". Half-lives (HLs) of Kis mRNAs are affected by blocking Xrn1 import, indicating that the nuclear function of Xrn1 is indispensable for the execution of the Kis pathway. We propose that the decay pathway of Kis mRNAs begins in the nucleus by Xrn1 binding to these mRNAs. Thus, our study unveils a nuclear step of the cytoplasmic mRNA decay pathway. The dependence of mRNA synthesis and decay on Xrn1 import is not binary. Instead, each gene is partially responsive to the Xrn1 import, the extent of which is defined as a "Kis value". Under optimal proliferation conditions, we observed a full spectrum of Kis values. Our model proposes that many mRNAs are exported to the cytoplasm while already carrying some of the degradation factors with them. This suggests that nuclear factors, including the transcription apparatus, are capable of regulating cytoplasmic mRNA decay. Importantly, the Kis pathway is important for the capacity of gene expression to properly respond to environmental changes.

## Results

### Xrn1 contains two functional nuclear localization sequences (NLSs)

To study Xrn1 shuttling, we first used cNLS mapper[18–20] and NLStradamus[21] programs. We found two potential NLSs in Xrn1, but not in other DFs. NLS1 of Xrn1 is a classical monopartite NLS (368-LEGERKRQRVGK-381), located at the junction between the "core" enzyme and a "linker domain"[22]. This linker, whose function is unknown, protrudes outside of the core and will be referred herein as "Tail" (Fig. 1a). The NLS1 sequence and its location within the Xrn1 structure are conserved in different yeast species (Fig. 1b). The presence of an NLS in a tail-like protrusion of Xrn1 was also found in Xrn1 homologues in higher eukaryotes (Supplementary Fig. 1a-b). NLS2 (1246-KKALEKKK-1255), which is located in the C-terminus (Fig. 1a), does not show significant sequence conservation; nevertheless, basic NLSs are present around the same amino-acid position in different yeast species (Supplementary Fig. 1c). Both NLSs are located in intrinsically disordered regions (Supplementary Fig. 1d). Using an import assay[3,23], we found that both NLSs are functional, as replacing the six basic residues of NLS1 with alanine (ΔNLS1) compromised Xrn1-GFP import (Fig. 1c and Supplementary Fig. 1e); Likewise, replacing four basic residues of NLS2 with alanine (ΔNLS2) also decreased import (Fig. 1c and Supplementary Fig. 1e). Import of Xrn1-GFP carrying the amino acid substitutions in both NLSs (Xrn1$^{\Delta NLS1/2}$-GFP) was almost entirely blocked (Fig. 1c, Supplementary Fig. 1e) while that of a co-expressed Pab1-RFP was unaffected by the mutations (Fig. 1d).

We have previously reported that several mRNA decay factors (DFs) are imported to the nucleus and function together in transcription regulation, probably as a complex[2,3]. To determine whether import of Pat1, Dcp2 and Lsm1 is dependent on Xrn1, we took advantage of a previous observation that during nutrient starvation they are found in the nucleus[3]. Indeed, amino acid substitutions in either NLS1 or NLS2 or both compromised import of these DFs (Fig. 1e-g and Supplementary Fig. 1f-h). Moreover affinity purification of Kap120-TAP pulled down both Xrn1 and Pat1 and Dhh1 in an NLS1/2-dependent manner (see below). Thus, Xrn1 NLSs are involved, directly or indirectly, in importing also other DFs.

### Amino acid substitutions that cause constitutive nuclear localization of Xrn1 tail are mapped to an R3H-like RNA binding motif

To further study NLS1 function outside the context of Xrn1 and NLS2, we replaced the Xrn1 core sequences with GFP, creating a "Tail-GFP"

fusion protein. Unexpectedly, despite the presence of NLS1, GFP-tagged Tail localized in the cytoplasm (Fig. 2a). However, using a standard shuttling assay, we discovered that the Tail-GFP can be imported (Fig. 2b), indicating that Tail-GFP shuttles between the two compartments. We suggest that its apparent exclusion from the nucleus is due to its slow import relative to its export kinetics.

After randomly mutagenizing Tail, a number of mutants were identified that localize almost exclusively in the nucleus (except for R507G) (Fig.2c-e, quantification of the nuclear/whole-cell ratio of the mean fluorescence intensity is discussed below). Thus, these mutations either compromised Tail export or enhance its import kinetics or both. Most of the single mutations map to a structural motif, that resembles a known RNA-binding motif R3H (Supplementary Fig. 2a-d). The RNA binding potential of the tail is also supported by a cryo-EM structure of Xrn1 bound to ribosomal RNA[24]. Tail$^{S454P}$-GFP is one of the constitutively nuclear mutants (Fig. 2c), but disruption of its NLS1 blocked its import (Fig. 2f), indicating that NLS1 is necessary for importing Tail-GFP. Unlike Rpb7-RFP, a shuttling protein[25] that was served as an internal control, Tail$^{S454P}$-GFP remained in the nucleus upon heat inactivation of the nucleoporin Nup49-313 and suppression of protein import (Fig. 2g-h)[26], suggesting that this mutation compromised Tail export.

### Tail binds RNA via the R3H-like motif

To directly determine whether Tail binds RNA, as was suggested by the presence of the R3H-like motif, we affinity purified an HTP-tagged Tail from UV cross-linked cells, followed by RNA trimming and radioactively labelling. Denatured PAGE of the labeled immunoprecipitated (IPed) material confirmed that Tail can bind RNA when expressed outside Xrn1 context (Fig. 3a). RNA binding of Tail$^{S454P}$ and Tail$^{M457T}$, carrying substitutions that also caused constitutive nuclear localization (Fig. 2c and e), was weaker than the WT tail (Fig. 3b), indicating that at least two mutations that affect the shuttling feature of the tail and are located in the R3H-like motif also affect RNA-binding.

Since NLS1 is positively charged and closely located to the RNA-binding motif, we surmised that NLS1 may contribute to RNA binding as well, and hypothesized that RNA binding would mask NLS1 from recognition by potential importins. Indeed, RNA-binding capacity of Tail$^{\Delta NLS1}$ (carrying disrupted NLS1) was defective (Fig. 3b), indicating that NLS1 also contributes to the RNA binding feature of the tail. Compared to WT tail, S454P substitution reduced RNA binding by 7.2-fold while ΔNLS1 mutation weakened RNA binding by 4-fold. Strikingly, introduction of both S454P and NLS1 mutations weakened RNA binding by 22-fold (Fig. 3b), suggesting that NLS1 and the R3H-like motif cooperatively bind the same RNA. R507G and L398S substitutions, which are outside of the R3H-like motif, did not compromise Tail's capacity to bind RNA (Fig. 3b).

To determine if Tail binds RNA also in the context of the full-length Xrn1, we employed an Electrophoretic Mobility Shift Assay (EMSA; also named "gel-shift") and found that a single amino acid substitution in the R3H-like domain, S454P, was sufficient to affect RNA binding of full-length Xrn1(Fig. 3c). These results suggest that the Tail R3H-like motif is a major contributor to the RNA-binding activity by Xrn1. Previous work identified a critical arginine residue in the active site- R101- which interacts with the 5'phosphate of the decaying RNA substrate[3,27,28]. To examine whether the 5' end of the Tail-bound RNA binds the active site, we introduced this R101G substitution and found that it reduced the gel shift signal (Fig. 3c), indicating that the active site contributes to our gel-shift signal. Importantly, while the single substitution R101G or S454P decreased RNA-binding by 60 or 65%, respectively, Xrn1 bearing both R101G and S454P substitutions lost its RNA-binding capacity almost completely (Fig. 3c). This synergism suggests that the R101 and S454 bind the same RNA cooperatively. Being an exonuclease, Xrn1 must bind RNA directly, at least in its active site. Xrn1 also binds directly full-length RNA[29]. Other RNA-binding

motifs and binding features are unknown. As discussed above, we found that Tail binds RNA in conjunction with R101 of the active site. To further examine whether Tail is involved in the major RNA-binding activity by Xrn1, we determined the repertoire of RNAs that Tail binds in vivo, using the crosslinking and analysis of cDNA (CRAC) technique, and compared it to that of the Xrn1[WT]. We found a significant correlation between the RPKM values of mRNAs bound to Xrn1 and those that bound the plasmid-born Tail (Fig. 3d), indicating that Xrn1 and Tail bind similar repertoire of RNAs. The RNA-binding of the plasmid-derived Tail was correlated with the mRNA level (Supplementary Fig. 2e). Moreover, Tail binds equally well Kis and non-Kis mRNAs (Supplementary Fig. 2f. See Introduction for Kis definition). These results suggest that outside the context of full-length Xrn1, Tail binds RNA non-selectively. Taken together, the linkage between the active site and S454, and our observation that the tail binds the same repertoire of RNA as Xrn1 suggest that Tail provides a major RNA-

binding site for the substrate RNA whose 5' end reaches the active site – the decaying RNA. However, RNA binding by Tail itself appears not to be selective, implying that substrate selectivity arises from the full Xrn1 context or Xrn1 localization.

## The importin Kap120 is involved in nuclear import of decay-factors

Nuclear proteins are imported from the cytoplasm by importins. There are more than a dozen known importins in yeast. For identifying Xrn1 importins, we screened all known importins by co-IP, comparing Xrn1 WT and Xrn1[ΔNLS1/2], and singled out Kap120 (see Methods). Kap120 imports Rpf1 and Swi6 by binding their monopartite NLSs[30]. To examine whether Kap120-Xrn1 interaction is direct, we affinity purified recombinant 6xHis-tagged Kap120 from *E. coli*, and FLAG-tagged Xrn1, or its mutant derivatives, from yeast extract (the purity of Xrn1 can be evaluated in Fig. 3c, middle panel). We reacted the two proteins in vitro and

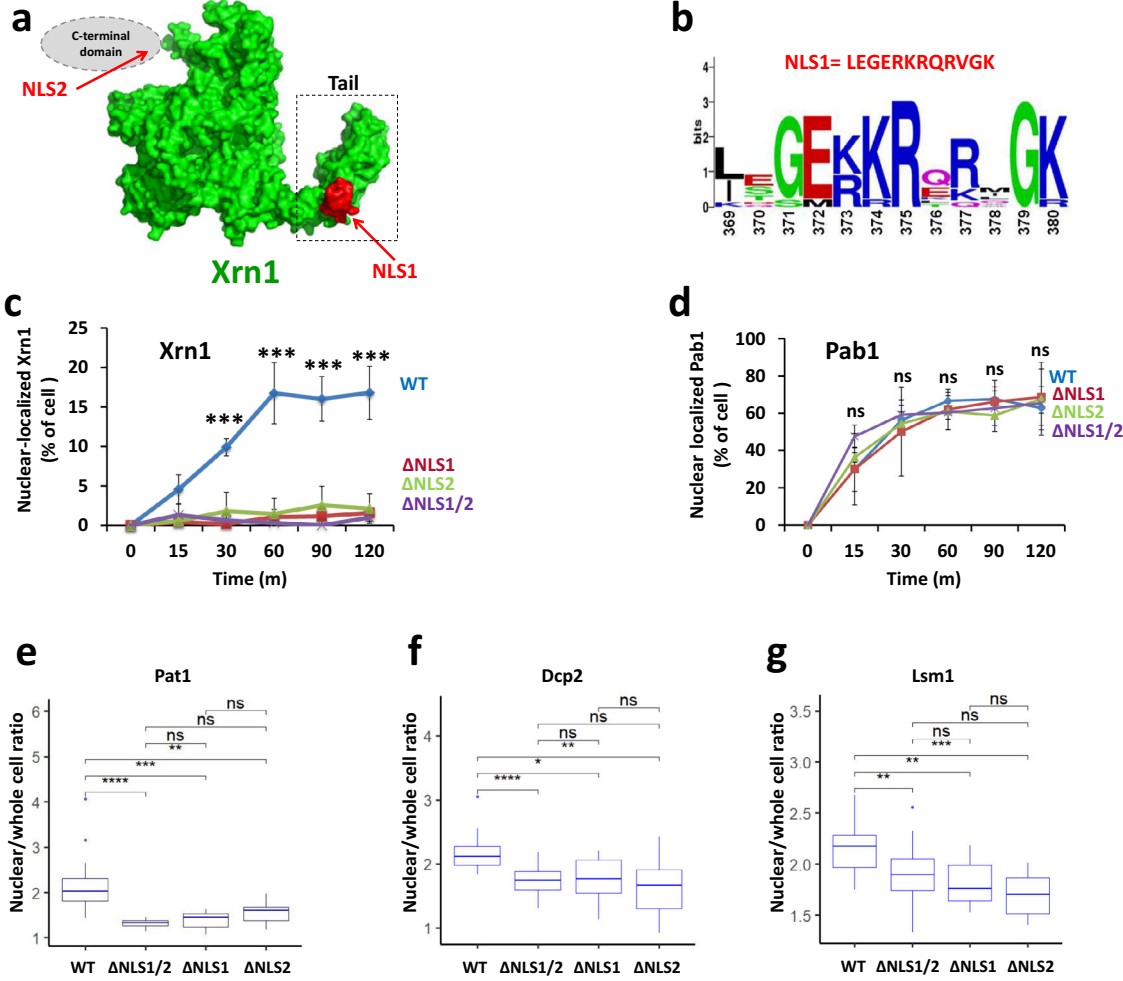

**Fig. 1 | Xrn1 contains two functional NLSs. a** *Position of NLSs*. NLS1, found by cNLSmapper[18–20], is highlighted in red. Xrn1 structure was modelled using SWISS-MODEL[63]. Tail is indicated by a dashed box. The expected location of the unstructured C-terminal domain containing NLS2 is shown schematically as a grey ellipse. **b** *Sequence conservation of NLS1 among different yeast species*. The sequences of 11 yeast strains were compared by the online WebLogo (U. Berkeley). The height of each stack of letters represents the degree of sequence conservation, measured in bits. **c** *Xrn1-GFP import kinetics*. A standard import assay was performed, as described in Methods, using cells that co-expressed Pab1-RFP (served as positive control, see d) and Xrn1-GFP or its mutant derivatives, as indicated. Percentage of cells showing nuclear import was measured at the indicated time intervals after inactivation of protein export. *n* = >100 cells examined over 3 independent experiments. Error bars represent standard deviation (S.D.) p-values of

two-tailed t-tests between WT and each of the NLS-mutant are indicated as *** <0.001 for all the pairs. **d** *Import of Pab1, used as a positive control*. Error bars in c and d represent standard deviation (S.D.) of three independent assays. **e–g** *Starvation-induced nuclear localization of the indicated fluorescent DFs is dependent on Xrn1 NLSs*. The mean fluorescence intensity of the nucleus and whole cell of WT or *NLS mutant* cells expressing the indicated RFP fused DF were measured after 1 h starvation in medium lacking sugar and amino acids. Each box represents the 25th to 75th percentile of values, with the median noted by the horizontal bar. Whiskers terminate at maxima/minima or a distance of 1.5 times the IQR away from the upper/lower quartile, whichever is closer. *n* = >300 cells examined over 3 independent experiments. The ratio between the nuclear and whole-cell fluorescence is shown as a boxplot. *P* value was calculated by Wilcoxon rank sum test. * - <0.05; **- <0.01; *** - >0.001; **** - <0.00001; ns – not significant.

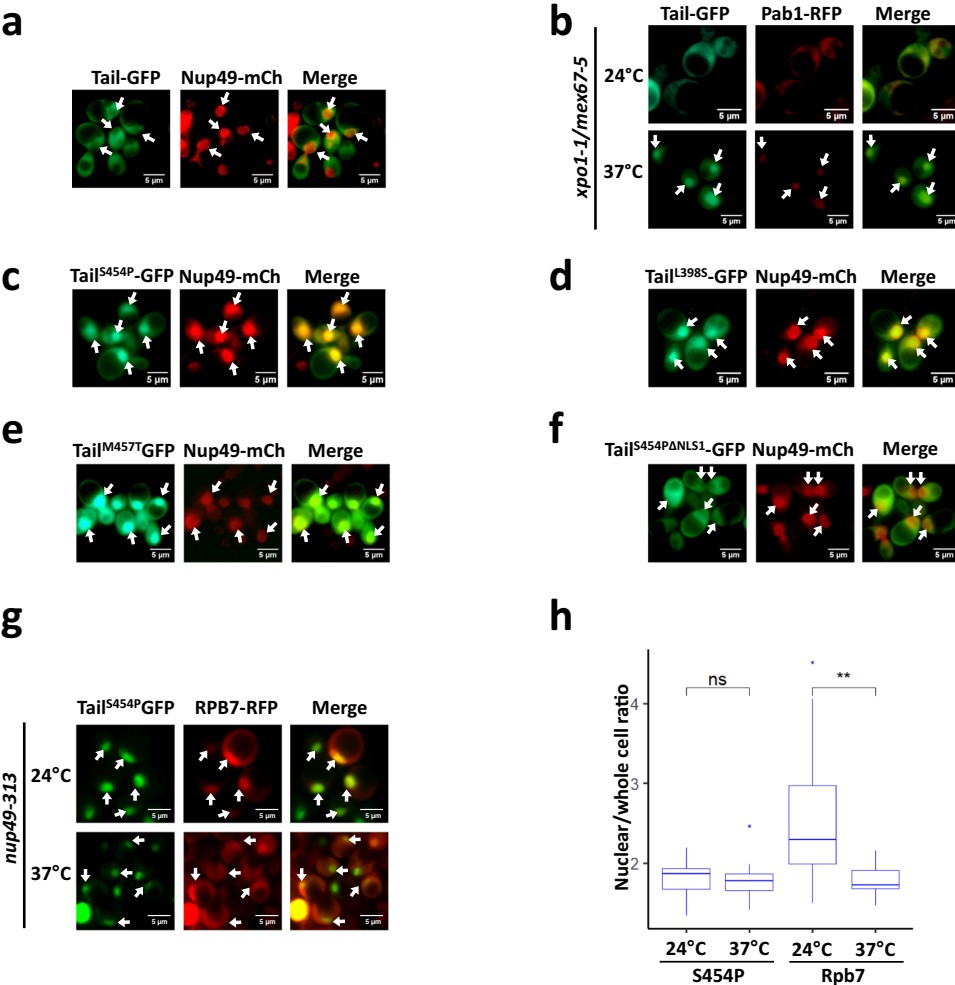

**Fig. 2 | Some mutant Tail-GFP molecules are localized constitutively in the nucleus.** Tail (amino acid 351 until 521 of the *XRN1* sequence) was fused to GFP and was expressed under *ADH1* promoter and terminator. Nup49-RFP (**a** nucleoporin) was used as a nuclear marker. **a** *During optimal proliferation, Tail is mainly cytoplasmic.* **b** *Tail-GFP shuttles between cytoplasm and nucleus.* Tail-GFP localization was examined by the import assay, as in Fig. 1c. Photos were taken after 2 h at 37 °C. **c-e** *Mutant Tail-GFP molecules that are constitutively localized to the nucleus.* The Tail sequence of Tail-GFP was randomly mutagenized by PCR mutagenesis protocol and introduced into WT strain by transformation. Distinct colonies were allowed to proliferate in 96 wells. The cellular localization of the mutants was automatically scanned by fluorescence microscopy using a robot. Shown are some of the mutants that were localized to the nucleus in optimally proliferating cells. See Fig. S2d for positions of the mutations. **f** *Tail^S454P nuclear import is mediated by NLS1.* NLS1 of Tail^S454P was mutagenized (resulting in Tail^S454P, ΔNLS1), and the mutant was analysed as in **c**. **g** *Localization of mutant Tail-GFP in temperature-sensitive (ts) nuclear-import mutant.* A shuttling assay, using cells that express *nup49-313* and co-expressed *xrn1^S454P* and *RPB7*-RFP (as a positive control), was performed as described previously[25]. **h** *Quantification of the shuttling assay.* The nuclear/whole-cell ratio of the mean fluorescence intensity was determined by ImageJ, as described in Methods. Each box represents the 25th to 75th percentile of values, with the median noted by the horizontal bar. Whiskers terminate at maxima/minima or a distance of 1.5 times the IQR away from the upper/lower quartile, whichever is closer. *n* = >100 cells examined over 3 independent experiments. *P*-value was calculated by Wilcoxon rank sum test. ** - *p* < 0.01. ns – not significant.

affinity purified the resulting complex, revealing that Xrn1 and Kap120 can interact in vitro and that Xrn1^ΔNLS1/2-Kap120 interaction was 4.5-fold weaker than Xrn1^WT-Kap120 (Fig. 4a), indicating that either NLS1 or NLS2 or both NLSs contribute to the in vitro interaction. The reciprocal IP experiment led us to the same conclusion (Supplementary Fig. 3a). Kap120 bound Xrn1^ΔNLS2 better than Xrn1^ΔNLS1 (Fig. 4b, designated in the figures as "ΔNLS1" and "ΔNLS2" respectively), indicating that, in vitro, NLS1 binds Kap120 better than NLS2. Kap120 also binds Tail, outside the context of Xrn1, in an NLS1-dependent manner (Supplementary Fig. 3b). A co-IP experiment with yeast extract demonstrates that NLS1/2-mediated Kap120-Xrn1 interaction occur also in vivo (Fig. 4c). Using a similar co-IP experiment, we found that Dhh1 and Pat1, but not Ccr4, also interact with Kap120 in vivo and, interestingly, this interaction was also ~5-fold reduced in the Xrn1^ΔNLS1/2 cells (Fig. 4d).

We took advantage of the constitutive nuclear localization of Tail^S454P-GFP (Fig. 2c) and deleted *KAP120* to examine its role in Tail^S454P-GFP nuclear localization. Deletion of *KAP120* changed significantly the cellular distribution of Tail^S454P-GFP (Fig. 4e), indicating that Kap120 plays a key role in Tail import. However, lack of Kap120 did not abolish import completely, suggesting that other importins may contribute to importing Tail, as well.

As NLS1 binds RNA (Fig. 3b, ΔNLS1), we determined the effect of RNA on the interaction of Kap120 with Xrn1^ΔNLS2, carrying only NLS1, and found that it weakened the interaction (Fig. 4f, ΔNLS2 lanes). The RNA had little or no effect on the interaction of Kap120 with Xrn1^ΔNLS1 (Fig. 4f, ΔNLS1 lanes), suggesting that NLS2 does not bind RNA in a manner that disturbs the interaction with Kap120. To corroborate the impact of RNA on NLS1 interaction with Kap120, we focused on the Tail domain that contains NLS1. Consistently, RNA could outcompete Kap120-Tail interaction in a dose-dependent manner (Supplementary Fig. 3c). We next asked whether, conversely, Kap120 can outcompete the interaction of Xrn1 with the RNA. As shown in Supplementary Fig. 3d, Kap120 can prevent Xrn1-RNA interaction in a dose-dependent fashion. Collectively, our data indicate that Kap120 binds the two NLSs

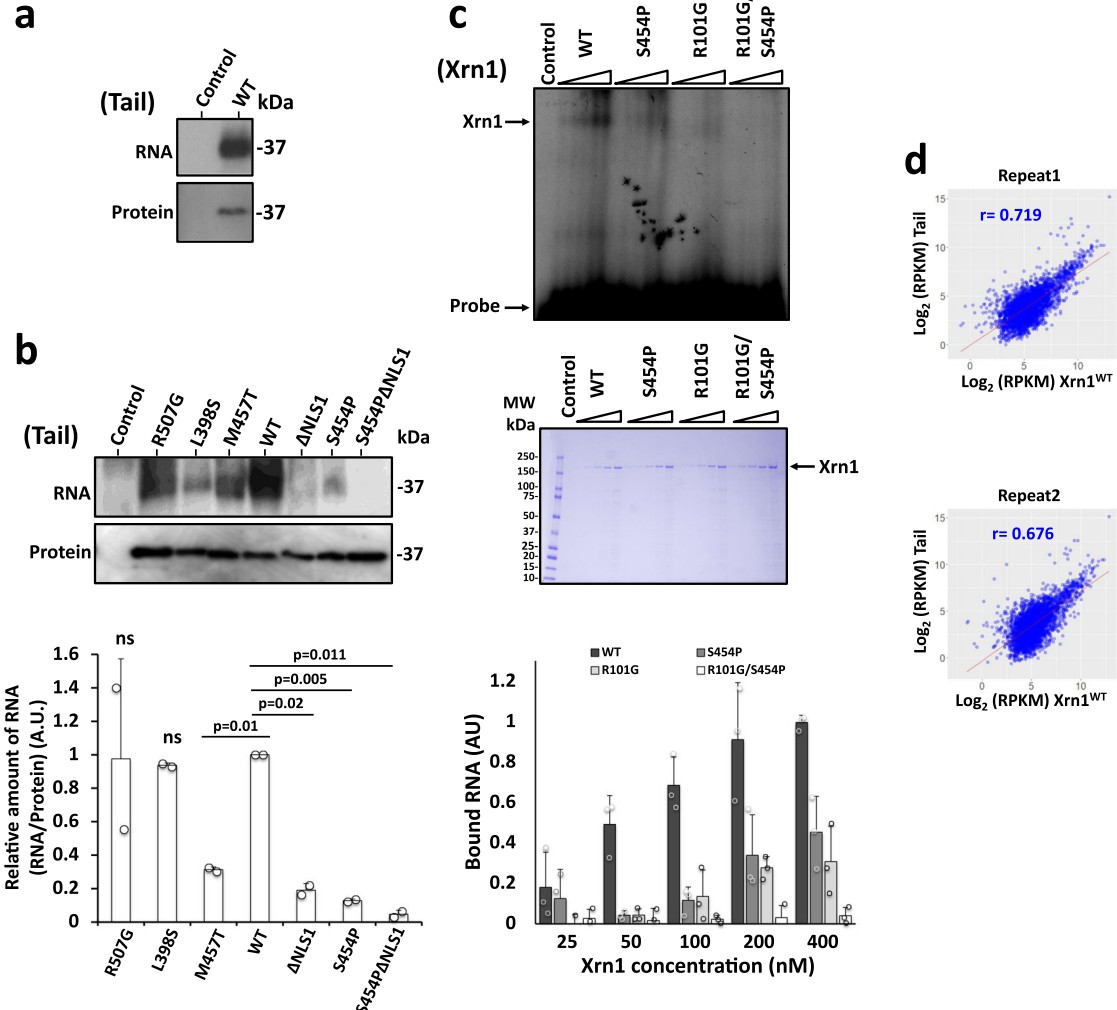

**Fig. 3 | Xrn1 Tail binds RNA. a** *Tail binds RNA*. Cells expressing 6xHis-TEV cleavage site-Protein A (HTP) tagged Tail, or control cells carrying no tag, were UV crosslinked. Tail-HTP-RNA complex was affinity purified, the co-purified RNAs trimmed and labelled with $^{32}$P and the protein-$^{32}$P-RNA complexes were separated in denaturing PAGE. Upper panel, autoradiogram ($^{32}$P-RNA); bottom panel, Western blot using anti-His antibodies. **b** *Some mutations in Tail (indicated above each lane) affect its RNA binding capacity.* Affinity purification of the indicated Tail mutants, which had been crosslinked to RNA, was performed as in A. The RNA and protein levels were quantified as described in Methods. RNA levels were normalized to the protein level. WT is arbitrarily defined as 1. Error bars represent standard deviation (S.D.) of 2 replicates. p-values were determined by two-tailed unpaired t-test. **c** *Tail provides a major RNA binding site of Xrn1 which acts in conjunction with the R101 residue of the active site*. FLAG-tagged Xrn1, or it's indicated mutant derivatives,

were affinity purified (see Methods), and subject to a gel-shift assay using $^{32}$P-5'-RNA (a mixture of 40, 32 and 28 b long), in the presence of EDTA to inactivate the enzyme (see Methods). Increasing amounts of purified proteins were used as indicated. Control lane represents experiment in which the probe was reacted with elute from untagged cells. Quantification of 3 biologically independent experiments is shown at the bottom. Error bars represent standard deviation (S.D.). The results of WT Xrn1 reacted with 400 μM was defined arbitrarily as 1. Purified Xrn1 was run in parallel to verify equal amount of protein loading. Shown is a Coomassie blue straining (middle panel). **d** *RNA bound by Tail is highly correlated to Xrn1 RNA interactome*. Tail and Xrn1 were subject to a full CRAC analysis. RPKM values of RNA binding to Tail and Xrn1 were scatter plotted. Two independent biological replicates are shown. Pearson's correlation (r) is indicated.

of Xrn1 and that Kap120 and the RNA compete for binding to Xrn1, while NLS2 binding to Kap120 does not appear to be mediated by RNA.

## Disruption of Xrn1 NLSs, or Kap120, affects both mRNA synthesis and decay rates

Identifying Xrn1 NLSs and the importin that recognizes them was then utilized to determine the effect of preventing Xrn1 shuttling on gene expression. We first deleted *KAP120* from WT cells (expressing WT Xrn1) and observed a significant drop in transcription (Fig. 4g and Supplementary Fig. 3e; Supplementary Data 3). Since Xrn1 is required in the nucleus for efficient transcription, these results were expected and are consistent with Kap120 being a major Xrn1 importin. Surprisingly, however, we also observed an effect on mRNA half-lives (HLs), even though Xrn1-mediated degradation occurs in the cytoplasm (Fig. 4g and Supplementary Fig. 3g). Although both mRNA synthesis

and decay were affected, the median mRNA steady state level was not changed (Fig. 4g and Supplementary 3f), indicating that the effect on mRNA synthesis was balanced with that on mRNA decay.

Since the full repertoire of Kap120 cargos is unknown, the effect of *KAP120* deletion on mRNA synthesis and decay kinetics cannot be directly attributed to lack of Xrn1 import. To more specifically address the role of Xrn1 import in regulating mRNA synthesis and decay rates, we took advantage of the mutations in NLS1/2 in cells that are otherwise WT. NLS1/2 mutations provide a powerful tool to study the possible linkage between mRNA degradation, import and transcription. Importantly, our mutations in Xrn1 NLSs neither change Xrn1 protein level (Supplementary Fig. 4a) nor the overall growth-rate (Supplementary Fig. 4b), and they do not affect Xrn1 enzymatic activity (see later).

As predicted, blocking import by mutating both NLS1 and NLS2 (using the *xrn1*$^{\Delta NLS1/2}$ strain) compromised the transcription rates (TRs)

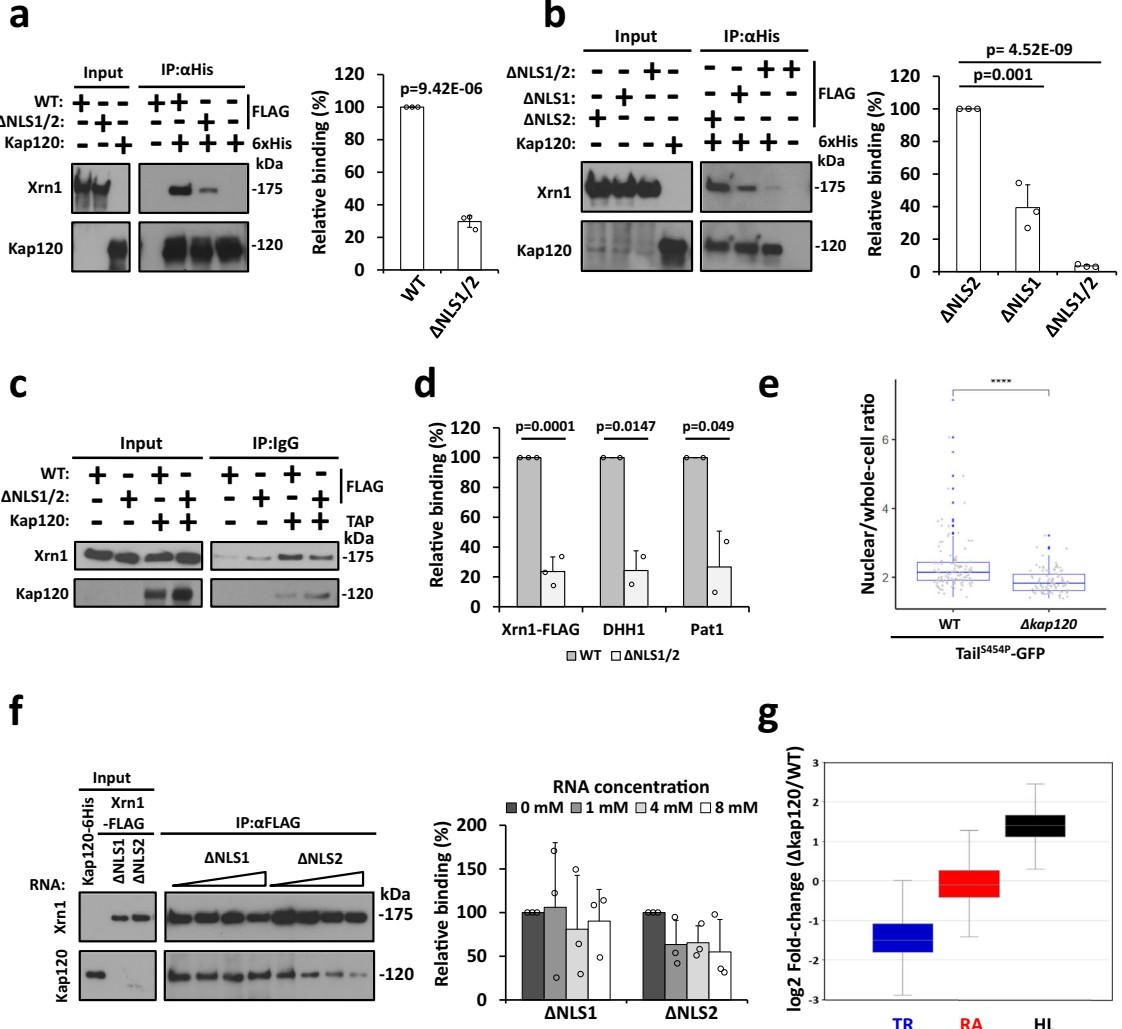

**Fig. 4 | Kap120 recognizes Xrn1 NLSs. a** *Interaction of Kap120 with Xrn1 is mediated by its NLSs.* Affinity purified FLAG-tagged Xrn1 or its mutant derivative, Xrn1[ΔNLS1/2], and recombinant Kap120-6xHis that had been purified with Ni-NTA column were mixed together and co-IPed with Ni-NTA column, followed by Western blotting. Xrn1-FLAG intensity was normalized to Kap120 intensity, defining Xrn1[WT]/Kap120 as 100%. $n = 3$ biologically independent experiments. Error bars represent standard deviation (S.D.). *p* values were determined by two-tailed unpaired t-test. **b** *Kap120 binds both Xrn1 NLSs.* Experiment shown in **a** was repeated three times. Xrn1-FLAG intensity was normalized to Kap120 intensity, defining arbitrarily Xrn1[ΔNLS2]/Kap120 as 100%. $n = 3$ biologically independent experiments. Error bars represent standard deviation (S.D.). p-values were determined by two-tailed unpaired t-test. **c** *Xrn1 interaction with Kap120 is mediated by NLSs in optimally proliferating cells.* TAP-tagged Kap120 was affinity purified with IgG-sepharose and the co-IPed proteins were subjected to Western blotting. Quantification is shown in **d**. **d** *NLS1/2 are required for efficient co-IP with Kap120 of Xrn1 and other DFs.* The membrane shown in **c** and two more membranes of additional replicates were also decorated with anti-Pat1 and anti-Dhh1 antibodies. Signals (minus that of the no-tag control) were normalized to that of Kap120. The normalized WT signal was defined as 100%. $n = 3$ (for Xrn1), $n = 2$ (for Dhh1 and Pat1) biologically independent experiments. Error bars represent standard deviation (S.D.). *p* values were determined by two-tailed unpaired t-test. **e** *Kap120 is used for efficient import of Tail[S454P]-GFP.* Optimally proliferating WT or Δ*kap120* cells expressing Tail[S454P]-

GFP and *NUP49*-mCherry (to mark the nucleus) were inspected microscopically. The nuclear/whole-cell ratio of the fluorescent signal was determined by ImageJ, see Methods. The mean ratio (Mean nuclear intensity/Mean whole-cell intensity) is represented in a jittered box-plot. Each box represents the 25th to 75th percentile of values, with the median noted by the horizontal bar. Whiskers terminate at maxima/minima or a distance of 1.5 times the inter quartile range (IQR) away from the upper/lower quartile, whichever is closer. The *p*-value was calculated using Wilcoxon rank sum test (**** - ≤ 0.0001) **f** *RNA blocks Xrn1-Kap120 interaction.* Interaction between purified Kap120-6xHis and FLAG-tagged Xrn1 carrying mutations in indicated NLS, was determine as in **a**, except that EDTA (to inactivate Xrn1) and increasing amounts of 40 b RNA were included. Xrn1-Kap120 complexes were affinity purified by anti-FLAG antibodies and analysed and quantified as in a. $n = >100$ cells examined over 3 independent experiments. Error bars represent standard deviation (S.D.). **g** *Deletion of KAP120 affects mRNA synthesis and decay, but not mRNA level.* Genomic Run-On (GRO) analysis was performed ($n = 3$), as described in Method. Box and whisker plot of the median levels (in arbitrary units) of *synthesis rate (SR), half-lives (HLs) and mRNA abundance (RA).* Each box represents the 25th to 75th percentile of values, with the median noted by the horizontal bar. Whiskers terminate at maxima/minima or a distance of 1.5 times the IQR away from the upper/lower quartile, whichever is closer. P value was calculated by Wilcoxon rank sum test ($p < 0.001$).

of >3000 genes (Fig. 5a, Supplementary Data 3). Note that the median decrease in TRs of *xrn1*[ΔNLS1/2] cells is similar to that in *kap120Δ* cells. The *xrn1*[ΔNLS1] strain is similarly defective in TR (Fig. 5a), albeit more modestly than the defect of *xrn1*[ΔNLS1/2] strain.

Consistent with the effect of *KAP120* deletion, not only TRs were altered in the mutants, but also mRNA half-lives (HLs). More than 2600

mRNAs became ≥2-fold more stable in the *xrn1*[ΔNLS1/2] mutant compared to WT (Supplementary Data 3). The median mRNA HLs in *xrn1*[ΔNLS1/2] was 2.5-fold higher than that in the WT strain (Fig. 5b). We observed highly significant overlap between the mRNAs whose both synthesis and decay were ≥ 2-fold affected by blocking Xrn1 import (Supplementary Fig. 4c). We named these 2401 overlapping mRNAs "Kem1 import

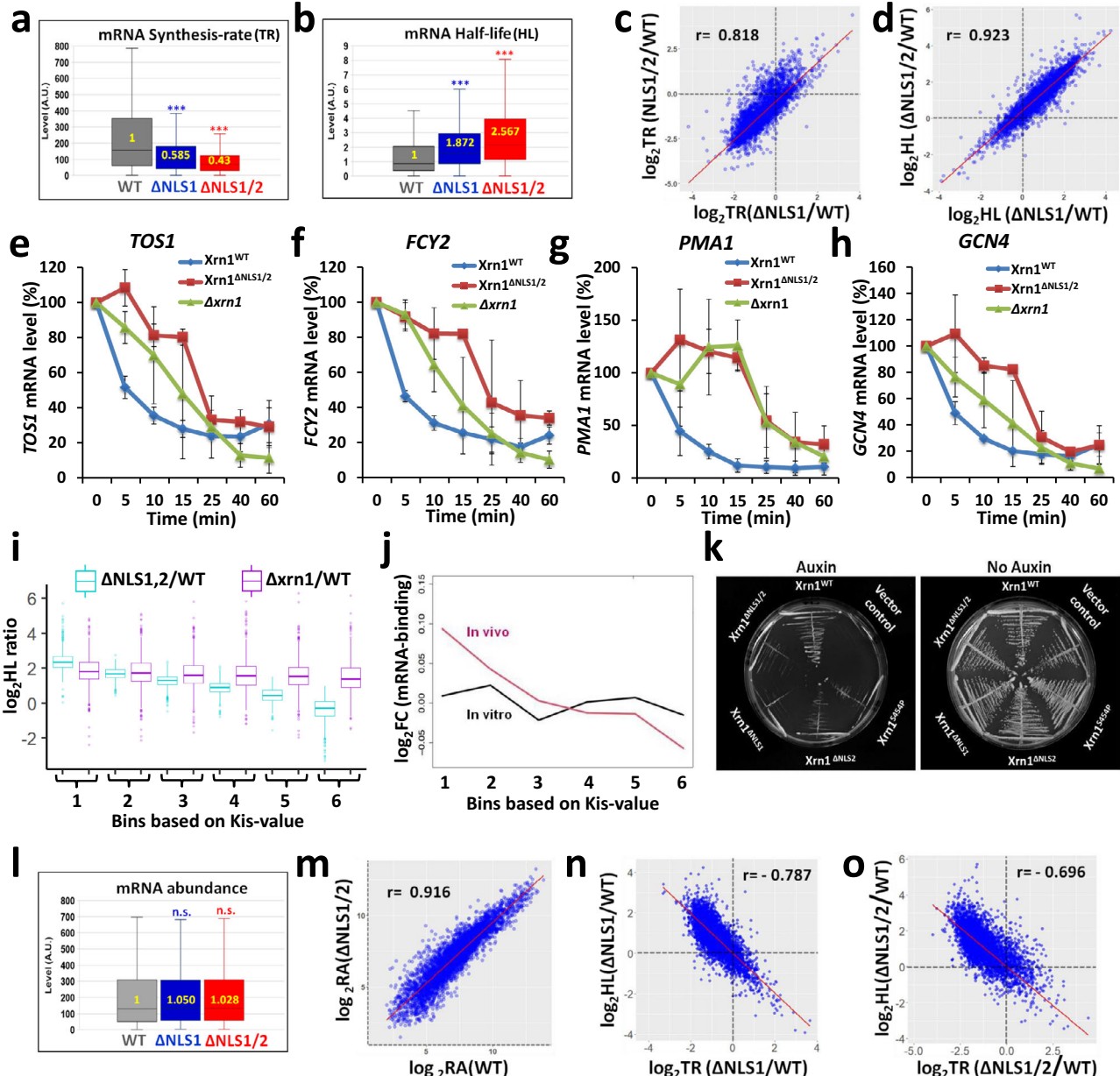

sensitive" ("Kis") mRNAs (Kem1 is an alias of Xrn1). Note that the median decrease in HLs of $xrn1^{\Delta NLS1/2}$ cells is similar to that in $kap120\Delta$ cells. These data indicate that nuclear import of Xrn1 is required for both mRNA synthesis and decay.

The effect of ΔNLS1 on TR and HL was correlated with the effect of ΔNLS1/2 on these parameters (Fig. 5c, d, respectively), consistent with the similar inhibition of Xrn1 import caused by ΔNLS1 and by ΔNLS2 (Fig. 1c). This strong correlation also suggests that disruption of NLS2 in $xrn1^{\Delta NLS1}$ cells does not change much the affected target genes.

A more direct experiment that determines HLs was performed by blocking transcription and monitoring the levels of specific mRNAs following the transcription arrest. A number of mRNAs became more stable in $xrn1^{\Delta NLS1/2}$ (Fig. 5e-h) as well as in strains carrying a single NLS disruption (Supplementary Fig. 4d-g). Collectively, this lack of a clear difference between ΔNLS1 and ΔNLS2, and the correlation between the strains (Fig. 5c, d), as well as the similar effect of ΔNLS1 and ΔNLS2 on Xrn1 import (Fig. 1c), suggest that the two NLSs function cooperatively in Xrn1 import and disruption of either one of them inhibits import (Fig. 1c) and affects the same repertoire of target genes.

These results prompted us to compare the effect on HLs of $xrn1^{\Delta NLS1/2}$ with that of complete deletion of *XRN1*. Strikingly, we found no difference in the median HLs ratio of Kis mRNAs in $xrn1^{\Delta NLS1/2}$ as compared with *Δxrn1* cells (Supplementary Fig. 4h, "Kis mRNA"). In contrast, HLs of the non-Kis mRNAs were affected by *XRN1* deletion, as expected, but very little affected by blocking Xrn1 import (Supplementary Fig. 4h, "non-Kis mRNA"). Thus, for Kis mRNAs, preventing Xrn1 import is analogous to a complete Xrn1 deletion, whereas non-Kis mRNAs are targeted for degradation by Xrn1ΔNLS1/2 almost as WT Xrn1. The two groups can be distinguished by the Gene Ontology (GO) terms of their products (Supplementary Fig. 4j), most of which are related to ribosome biogenesis and composition, and the length of their open reading frame (ORF) (Supplementary Fig. 4k). Importantly, the efficient decay of non-Kis mRNAs in $xrn1^{\Delta NLS1/2}$ cells indicates that the enzymatic activity of Xrn1ΔNLS1/2 is not affected by the mutations in the NLSs. Note that the median HLs of the Kis and non-Kis mRNAs in *xrn1Δ* cells is comparable (Supplementary Fig. 4h), indicating that the decay of both groups similarly requires Xrn1; the distinction between them is apparent only upon blocking Xrn1 shuttling. We also found no correlation between the effect of ΔNLS1/2 on HL-ratio and the actual HL

**Fig. 5 | During optimal proliferation conditions, blocking Xrn1 import leads to inefficient mRNA synthesis and decay, without affecting mRNA levels.** GRO analysis was performed, in three replicates, as described in Method. Box and whisker plot of the median levels (in arbitrary units) of **a** *transcription rate (TR)*; **b** *half-lives (HLs)*. Each box represents the 25th to 75th percentile of values, with the median noted by the horizontal bar. Whiskers terminate at maxima/minima or a distance of 1.5 times the IQR away from the upper/lower quartile, whichever is closer. p-values were obtained by Wilcoxon rank sum test, ***: p < 0.001. **c** *Defect in TR of xrn1^ΔNLS1 is correlated with that of xrn1^ΔNLS1/2*. Each spot represents an average of 3 replicates. Pearson correlation coefficient (r) are indicated. **d** *Defects of mRNA decay (expressed as in C) of xrn1^ΔNLS1 is correlated with that of xrn1^ΔNLS1/2*. Each spot represents an average of 3 replicates. Pearson correlation coefficient (r) are indicated. **e-h** *The effect of mutating NLSs on decay rates of specific mRNAs, determined by Northern blot hybridization*. Decay assay described in Methods. Shown are mRNA levels, quantified using the PhosphorImager technology and normalized to *SCR1* mRNA, as a function of time post-transcription arrest. n = 2 biologically independent experiments. Error bars represent standard deviation (S.D.). **i** *For mRNAs with low Kis-values (bins 1-3), the effect of xrn1^ΔNLS1/2 mutations on HLs is comparable to Δxrn1*. mRNAs were arranged in a ascending order based on their kis-values and divided into 6 bins with equal number of genes. The ratios between HL in the mutant and that in WT is shown. Box and whisker plot of the log₂ HLs ratios are shown. N = 3 biological replicates. Each box represents the 25th to 75th percentile of values, with the median noted by the horizontal bar. Whiskers terminate at maxima/minima or a distance of 1.5 times the IQR away from the upper/lower quartile, whichever is closer. Note that mRNAs in most bins were affected by *XRN1*

deletion. **j** *Xrn1 and Xrn1^ΔNLS1/2 equally bind mRNA in vitro, as opposed to differential binding in vivo*. In vivo or in vitro mRNA interactome profiles of WT and ΔNLS1/2 mutant forms of Xrn1-FLAG were determined, each condition in a duplicate, as follows. Condition I: in vivo. Live cells were UV cross linked followed by a standard RIP-seq. Condition II: in vitro. Purified Xrn1 samples were reacted in vitro with affinity purified poly(A) + mRNA followed by RIP-seq. For each condition, the log2 fold change (FC) of mutant to WT RIP-seq reads was computed, and the median taken within each of these predefined bins (see **i**). Centering was performed computationally. **k** *Normal proliferation of cells carrying mutations in Xrn1 NLSs is dependent on SKI2*. Synthetic sickness assay of NLSs mutants with *ski2Δ*. *ski2Δ*, *xrn1*-AID strains carrying the indicated plasmid were streaked on plates lacking (right panel) or carrying Auxin (that induced Xrn1-AID degradation). Photos were taken after 2 days at 30 °C. **l** *During optimal conditions, mRNA abundance is unaffected by mutations in Xrn1 NLSs*. mRNA abundance was determined (Methods). Each box represents the 25th to 75th percentile of values, with the median noted by the horizontal bar. Whiskers terminate at maxima/minima or a distance of 1.5 times the IQR away from the upper/lower quartile, whichever is closer. p-values were obtained by Wilcoxon rank sum test and were found to be not significant (n.s.). Median values in arbitrary unit (A.U.) are indicated inside the plots. **m** *mRNA Abundance (RA) in xrn1^ΔNLS1/2 are correlated with those in WT*. Each spot represents an average of 3 biological replicates. Pearson correlation coefficient (r) is indicated. **n-o** *The effects of NLS1 (n) or NLS1/2 (o) disruption on transcription rates (TR) are correlated with their effects on HLs*. Scatter-plot of log₂ mutant/WT ratio. Each spot represents an average of 3 replicates. Pearson correlation coefficient (r) is indicated.

---

values in WT cells (Supplementary Fig. 4L), indicating that the distinction between Kis and non-Kis is not based on mRNA stability.

Biological processes are often not subject to a simple "on/off" regulation. We surmised that mRNAs are degraded by more than just one pathway, and that the relative contribution of the different pathways is mRNA specific. We have thus assigned a "Kis value" to each mRNA, which represents the sum of the following two figures: 1. HL ratio (ΔNLS1/2 HL/WT HL) and 2. The inverse TR ratios (WT TR/ΔNLS1/2 TR). Thus, the Kis value represents the overall impact of Xrn1 import on the two kinetic features that determine mRNA level. Note that because the effects of Xrn1 import on these features are inversely correlated (it negatively affects TR but positively affects HL), the effect on one kinetic feature can outbalance the effect on the other. We next divided the mRNAs into equal bins, based on their Kis value (Supplementary Data 4), whereby bin 1 represents most affected mRNAs, having the highest Kis values. Dividing Kis values into 11 bins demonstrated that the effect of the NLS1/2 mutations on the Kis values was almost perfectly gradual for both TRs and HLs (when plotted in Log₂ scale), despite an unbiased division to bins with equal number of genes (Supplementary Fig. 4i). This indicates that the effects of Xrn1 import on HL and TR are, indeed, not binary but represent full spectra. When the impact of *XRN1* deletion on HLs was examined on six equal bins (we used six bins for simplicity and better statistical power) we found similar mRNA stabilization across all bins (Fig. 5i, "Δxrn1/WT"), indicating that the decay of all six groups similarly requires Xrn1. Thus, the distinction between the groups is apparent only upon blocking Xrn1 shuttling (Fig. 5i, "ΔNLS1,2/WT"), indicating that it is the nuclear Xrn1 that is capable of affecting the Kis value.

A possible explanation for the effect of mutating NLS1/2 is that they compromise interactions with the substrate and/or other components of the cytoplasmic mRNA degradation machinery. This possibility is unlikely because deleting Kap120, in Xrn1 WT background, resulted in a similar mRNA destabilisation (Fig. 4 g). Nevertheless, to address these possibilities more directly, we first found that Xrn1^ΔNLS1/2 binds equally well a number of other mRNA decay factors (Supplementary Fig. 4m). We next examined whether the mutations in NLS1/2 affected the interaction with mRNAs. To this end, we determined the in vitro interaction of WT Xrn1 and Xrn1^ΔNLS1/2 mutant with purified cellular mRNAs. Our results demonstrate that the two forms of Xrn1 bound mRNAs equally well; equal binding was observed for all Kis

values (Fig. 5j, "in vitro"). This indicates that the mutations in both NLSs do not detectably affect Xrn1 binding to mRNAs. In parallel, we performed a similar in vivo analysis, i.e., UV cross-linking of live cell followed by purifying Xrn1 or Xrn1^ΔNLS1/2, which revealed a Kis valuesensitive differential interaction (Fig. 5j, "in vitro"). The difference between the in vitro and in vivo results demonstrates that the interaction of Xrn1 with its substrates in vivo is not a trivial consequence of its interaction capacity with its substrate in vitro. Taken together, the mutations in Xrn1 NLSs do not affect Xrn1 capacity to bind other DFs or its RNA substrates. Interestingly, preventing import increases binding of Xrn1 to mRNAs with high Kis values. We interpret this observation to indicate that Xrn1 can bind all mRNAs in the cytoplasm but "productive" cytoplasmic binding that results in degradation occurs only for mRNAs with low Kis values.

Cells require mRNA degradation activity for viability and cannot survive in the absence of the two major RNA exonucleases - Xrn1 or the exosome[31,32]. We introduced the auxin-induced degron (AID) tag[33] into *XRN1* (at its natural chromosomal locus) in cells lacking the exosome subunit *SKI2*. We used this system to verify the role of Xrn1 NLSs in mRNA decay by introducing plasmids expressing either *XRN1* or its NLS mutant derivatives or a vector control into this Δ*ski2, XRN1*-AID strain. In the absence of auxin, these Δ*ski2, XRN1*-AID cells grew normally (Fig. 5k, "No auxin"). Plating these cells on auxin-containing medium stimulated Xrn1-AID degradation whose sole source of Xrn1 was the plasmid-borne Xrn1. We found that a mutant form with disruption of a single NLS, or disruption of both NLSs, became sick when plated on a medium containing auxin (Fig. 5k, "Auxin"). The synthetic sickness of NLS mutants with Δ*ski2* is consistent with a defect of the NLSs mutants in mRNA decay. We propose that in case Xrn1 is not imported and Kis mRNAs cannot be degraded by Xrn1 (Supplementary Fig. 4h), the exosome takes over the role of Kis mRNA degradation.

## Under optimal proliferation conditions, Xrn1 import does not affect mRNA steady-state levels

Intriguingly, the median steady state (SS) level (Fig. 5 l), and the SS level of specific mRNAs (Fig. 5m) remained constant. Thus, under optimal environmental conditions, shuttling of Xrn1 affects the dynamics of mRNA biogenesis and turnover without changing overall mRNA levels. A significant correlation was found between the effect of blocking Xrn1 import, using *xrn1*^ΔNLS1 or *xrn1*^ΔNLS1/2, on TR and its effect on HL (Fig. 5n-

o). Thus, the more mRNA synthesis rate is dependent on Xrn1 import, the more its decay rate is affected. Collectively, the capacity of Xrn1 to degrade mRNAs is related to its capacity to stimulate their synthesis.

### Xrn1 nuclear import regulates exit from quiescence and proliferation in fluctuating (oscillating) temperatures

We have examined several environmental conditions (e.g., high salt, high osmolarity, high pH, low pH, high temperature, low temperature, ethanol or glycerol as the sole carbon source) and found out that, under many of these conditions, disruption of Xrn1 shuttling had little effect on the proliferation rate (in contrast with *XRN1* deletion). However, when we plated cells that had experienced long-term starvation (≥7 days), we observed that *xrn1*[ΔNLS1/2] colonies were smaller than WT, raising the possibility that they exited starvation abnormally slowly. A characteristic feature of yeast cells that exit the starvation is an increase in the proportion of budding cells. Indeed, the budding proportion of WT cells increased during exit, exhibiting two budding phases due to cell synchronization. In contrast, budding of *xrn1*[ΔNLS1/2] was delayed and unsynchronized (Fig. 6a). We also used flow cytometry to view the progression into the cell cycle. This assay showed that starved WT cells ("time 0") consist of 75% cells arrested with 1 copy of their genome (1 N, i.e., $G_0$ or G1) and virtually no S-phase cells, whereas *xrn1*[ΔNLS1/2] cells consist of 70% cells in $G_0$ or G1 and ~12% cells in S-phase. This suggests that the mutants did not arrest properly in $G_0$ and some cells stopped dividing before DNA replication was completed, or started replication during starvation. More importantly, by 3.75 h and to a greater extent at 7.5 h post re-feeding, the WT cells started to replicate their genome and exiting stationary phase, whereas the mutant cells exhibited abnormally delayed signs of exiting starvation (Fig. 6b, c and Supplementary Fig. 5a). In contrast, in optimal conditions when cells proliferated exponentially in rich medium, we observed no differences in DNA content of the two strains (Supplementary Fig. 5b).

These results provoked us to examine Xrn1 localization post-refeeding. We found that ~20% of starved cells accumulated Xrn1 in the nucleus. During the first phase of exit from starvation (the kinetics are dependent on the time that cells experience the starvation), cells continued to accumulate nuclear Xrn1 and the proportion of cells displaying nuclear localized Xrn1 gradually increased. Later, the nuclear localization of Xrn1 gradually decreased until it reached the baseline that characterizes optimally proliferating cells (Fig. 6d, e "WT", Supplementary Fig. 5c, "Xrn1[WT]-GFP" panels). As expected, disruption of NLS1/2 resulted in little nuclear localization (Fig. 6d and Supplementary Fig. 5c). Interestingly, NLS2 but not NLS1 was responsible for importing Xrn1 in response to re-feeding (Fig. 6e and Supplementary Fig. 5c). Xrn1[ΔNLS1]-GFP localization paralleled that of Xrn1-GFP until 3 h post re-feeding. Later, during entry into log phase, NLS1 became an important NLS (Fig. 6e, >3 h), as we observed in optimally proliferating cells (Fig. 1c). Thus, starvation-induced import of Xrn1 is specifically mediated by NLS2. Moreover, during the transition from starvation to rich medium, there is an additional wave of import that is specifically dependent on NLS2 as well. Not only was Xrn1 imported during the exit from starvation, but also Pat1 was imported concurrently with Xrn1 in an NLS1/2-dependent fashion (Supplementary Fig. 5d). Kap120 was involved in Xrn1 localization during starvation, as well as in the additional import observed during the first phase of exit from starvation that peaked at 2.25 h post-refeeding (Fig. 6f). Consistent with the phenotypes that characterized Xrn1[ΔNLS1] cells, discussed above, we found that Xrn1 import is required for efficient accumulation of a subset of mRNAs (Fig. 6g, h and Supplementary Fig. 5e) (116 mRNAs, Supplementary Data 5). Taken together, Xrn1 import is required for proper exit from stationary phase in response to re-feeding, which includes proper mRNA accumulation. During optimal proliferation, both NLS1 and NLS2 are required for efficient import, however, during long-term starvation and exit from it, NLS2 seems to play the major role.

The impact of Xrn1 import on the cell responses to changes in nutrient availability, which characterizes the transition from starvation to sated conditions, encouraged us to examine cell proliferation rate under continuous environmental changes. For simplicity we chose fluctuating temperatures. Exponentially proliferating cells were cultured in a thermal cycler, enabling us to oscillate the temperature between heat shock (38 °C for 150 min) and cold shock (8 °C for 150 min). Each of these temperatures, although extreme for yeast growth, permitted cell proliferation with no difference between the strains when used continuously (Supplementary Fig. 5f-h). However, when the cultures were fluctuated between these temperatures, *xrn1*[ΔNLS1/2] grew slower than WT cells (Fig. 6i). Collectively, we find that the shuttling feature of Xrn1 is not required for proliferation during continuous conditions of unchanging environment, however harsh; it is required mainly, if not exclusively, for proper and rapid responses to environmental changes.

## Discussion

The eukaryotic cell contains a nucleus and a cytoplasm, the linkage of which involves the transport of various molecules between these compartments. The importance of this linkage is exemplified here by the mechanisms that control mRNA levels, which are maintained by a balance between RNA synthesis and decay. We hypothesized that proper linkage between these two opposing processes involves shuttling of factors that promote mRNA synthesis and decay: transcription factors and mRNA decay factors. As the mRNA decay complex mediates both processes[8,9], we surmised that its shuttling is instrumental for the linkage between the nuclear mRNA synthesis and the cytoplasmic decay. Here we uncover a nucleocytoplasmic shuttling mechanism of Xrn1 and some of its associated factors (Pat1, Dcp2, Lsm1) that is regulated, in part, by the bound RNA, which in turn, regulates mRNA synthesis and decay.

Bioinformatic analyses revealed two NLSs in Xrn1, but none in a number of other known DFs that we examined. We verified that these NLS sequences are necessary for nuclear import and showed that they are recognized by the little-studied karyopherin Kap120. Indeed, deleting *KAP120* resulted in aberrant mRNA synthesis and decay (Fig. 4g). Although, in vitro, Kap120 binds NLS1 better than NLS2 (Fig. 4b), disruption of either NLS compromised Xrn1 import substantially (Fig. 1c) and resulted in a reduced mRNA decay rate (Supplementary Fig. 4d-g). Consistently, the effect of disruption of a single NLS on TRs and HLs is correlated with that of disruption of both NLSs (Fig. 5c, d). These data suggest that the two NLSs are cooperatively required for Xrn1 import and disruption of either inhibits import. Importantly, only NLS1 binds RNA, while NLS2 does not. Disruption of just NLS2, which abolishes Xrn1 import to the nucleus, affected mRNA decay (Supplementary Fig. 4d-g). Thus, preventing Xrn1 import by disrupting any NLS affects mRNA decay – despite normal RNA binding of the mutant enzyme and its normal enzymatic activity. This conclusion was demonstrated using two more direct approaches, as discussed below. The underlying mechanism of cooperativity remains to be determined. Are two NLSs needed to cover Xrn1 from its different ends to enable efficient transport through the nuclear pore, as was proposed in other cases of large proteins[34]?

In contrast to optimal proliferation conditions, upon starvation, the two NLSs seem to acquire more distinct functions. Disruption of NLS2, but not NLS1, inhibited Xrn1 import. This was observed both during long-term starvation (Fig. 6e time 0) and during exit from starvation, upon re-feeding (Fig. 6e), which induced an additional wave of Xrn1 import (Fig. 6d, e). We therefore suspect that the proposed cooperation between NLS1 and NLS2 does not operate under all environmental conditions and could be subjected to regulation.

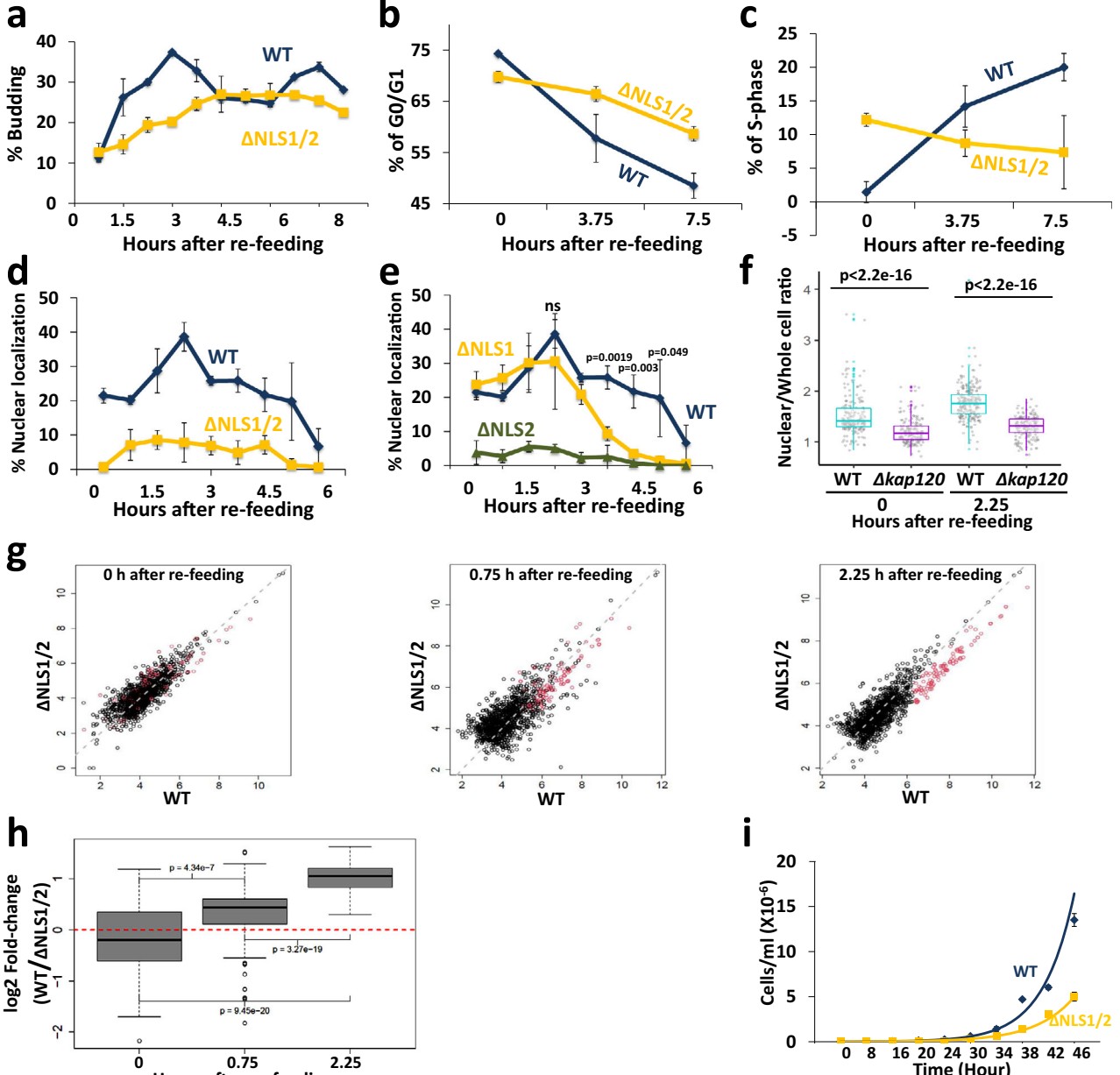

**Fig. 6 | Xrn1^ΔNLS1/2 abnormally responds to changes in the environment.**
**a** *xrn1^ΔNLS1/2 mutant cells bud abnormally slowly during exit from long-term starvation to sated conditions.* Seven days starved cells were re-fed (see Methods) and percentage of budded cells was plotted as a function of time post re-feeding. n = 2 biologically independent experiments. Error bars represent standard deviation (S.D.). **b**, **c** *xrn1^ΔNLS1/2 mutant cells enter abnormally slowly into S-phase, as assayed by* Flow cytometry (*FACS*) analysis. Cells were re-fed as in **a**. DNA content was determined by FACS, and plotted as a function of time post-re-feeding. *n* = 3 biologically independent experiments. Error bars represent standard deviation (S.D.). **d** *Xrn1^WT-GFP, but not Xrn1^ΔNLS1/2-GFP accumulates in the nucleus concomitantly with bud appearance.* Cells were inspected under a fluorescence microscope and % cells with nuclear localization of GFP was plotted as a function of time following re-feeding. Error bars represent S.D. of 3 replicates. Note that, in this import experiment, no cycloheximide was added to avoid its effect on the actual process of exiting the starvation. Therefore, in this case (unlike our other experiments), we cannot differentiate between import of pre-existing Xrn1 or newly synthesized one. *n* = 3 biologically independent experiments. Error bars represent standard deviation (S.D.). **e** *NLS2 is required for Xrn1 import during starvation and exit from starvation, whereas NLS1 is required only following 3 h post re-feeding.* The indicated mutant cells were examined following re-feeding as in **d**. *n* = 3 biologically independent experiments. Error bars represent standard deviation (S.D.). p-values were

determined by two-tailed unpaired t-test. ns not-significant. **f** *Kap120 is required for efficient import of Xrn1 during starvation and upon exit from starvation.* WT or Δ*kap120* cells, expressing *XRN1*-GFP and *RPB7*-RFP (to mark the nucleus), were inspected microscopically. After examining the starved cells (Time 0), the culture was re-fed as in **a** and examined microscopically at 2.25 h later. The nuclear/whole-cell ratio was performed as in Fig. 4e. P-value, based on 3 replicates, was calculated by Wilcoxon rank sum test. **g** *Xrn1 induces transcription of a group of genes during exit from starvation.* Starvation exit experiment was done as in **a**. RNA was sequenced (Methods). Reads were normalized to spike-in (S. pombe) and plotted (Methods). **h** *NLS1/2 of Xrn1 are necessary for efficient expression of a group of mRNAs during exit from starvation.* Genes that were sensitive to Xrn1 NLSs (red dots in g) were selected. Fold-change of mRNA level is expressed as a ratio between the normalized WT and the normalized ΔNLS1/2 mutant mRNA level. Each box represents the 25th to 75th percentile of values, with the median noted by the horizontal bar. Whiskers terminate at maxima/minima or a distance of 1.5 times the IQR away from the upper/lower quartile, whichever is closer; P-values from the Wilcoxon signed-rank test are indicated for each comparison. **i** *Xrn1 import is required for normal proliferation rate under fluctuating temperatures.* See Methods. Both strains proliferated comparably at constant temperatures (see Fig. S5d-f). *n* = 2 biologically independent experiments. Error bars represent standard deviation (S.D.).

It was previously reported that overexpressing Xrn1 and forcing it into the nucleus by tagging it with SV40 NLS suppressed the temperature sensitivity of the nuclear Xrn1 homolog, Rat1-1[35]. Evidently, however, the Xrn1 molecules that naturally import to the nucleus cannot suppress *rat1-1* temperature sensitivity, raising a seemingly paradox. However, there are some differences between the natural and the engineered Xrn1. (I) The natural level of Xrn1 in the nucleus is much lower than that of the engineered Xrn1[35]. (II) SV40 NLS is recognized by Kap95 (e.g[36]), whose Ran–GTP binding constant is 3 orders of magnitude lower than that of Kap120[37]. It was therefore proposed that, unlike Kap95 that releases its cargo as soon as it enters the nucleus and encounters Ran-GTP, Kap120 does not release its Xrn1 cargo immediately after nuclear entry[37], and potentially not until it reaches its destination – the chromatin[3]. Collectively, these quantitative, contextual and localization differences might provide a plausible explanation for the temperature sensitivity of *rat1-1* cells expressing WT *XRN1*.

The majority of yeast mRNAs are degraded in the cytoplasm in a pathway culminating by Xrn1 degradation[3,5,8]. Prior to this work, we hypothesized that preventing Xrn1 import would decrease mRNA synthesis of target genes and increase the decay of their transcripts because more Xrn1 would be available in the cytoplasm for mRNA decay. Our results proved otherwise. Specifically, cells carrying either mutations in NLS1/2 or deletion of *KAP120* exhibited abnormally slow decay rates, despite normal RNA binding capacity (Fig. 5J) and normal enzymatic activity of Xrn1 (see below) in these mutant cells. We propose that mRNAs stability is regulated, in part, by a decay pathway that initiates in the nucleus–the Kem1 import-sensitive (Kis) pathway (Fig. 7).

"Kis mRNAs" are defined as those whose stability is regulated mainly by the Kis pathway, i.e., mRNAs whose HLs increase >2-fold in NLS1/2 mutant. The decay rates of these mRNAs were comparable to those in Δ*xrn1* cells (Supplementary Fig. 4h). We therefore propose that, if Xrn1 is not imported, Kis mRNAs are degraded mainly by an Xrn1-independent pathway, e.g., the exosome (Fig. 7b, left pathway). Indeed, *xrn1*^ΔNLS1/2 cells become sick upon deletion of *SKI2*, encoding a cytoplasmic exosome component (Fig. 5k). These mRNAs are also degraded by the classical pathway, albeit inefficiently.

Degradation of non-Kis mRNAs is little dependent on Xrn1 shuttling (Supplementary Fig. 4h) suggesting that initiation and the entire execution of their decay pathway occurs in the cytoplasm. Normal decay of the latter mRNAs indicates that Xrn1^ΔNLS1/2 is enzymatically functional. This conclusion is supported by the WT growth rate of *xrn1*^ΔNLS1/2 cells under many different constant environmental conditions, except for fluctuating environment (see below).

Kis and non-Kis mRNAs can be distinguished by the effect that Xrn1 import has on (i) their TR (this was one of the criteria of their definition), (ii) the Gene Ontology (GO) terms of their products (Supplementary Fig. 4j); most of the Kis GO terms are related to ribosome biogenesis and composition, whereas the GO terms of non-Kis genes were different and exhibited little connectivity, and (iii) the length of their open reading frame (ORF) (Supplementary Fig. 4k). It is worth noting that the distinction between Kis and non-Kis mRNAs is unrelated to their HLs in WT cells (Supplementary Fig. 4l).

Although the distinction between Kis and non-Kis mRNAs is practically convenient, in reality, the dependence of mRNA synthesis and decay on Xrn1 import is not binary. Instead, each gene is partially responsive to the Xrn1 import, extent of which is defined as a "Kis value" (the higher the value – the higher the response). Under optimal proliferation conditions, we observed a full spectrum of Kis values (Supplementary Fig. 4i). Under these conditions, the spectra of TRs and HLs are reciprocal to each other (Supplementary Fig. 4i), illustrating the balance between them – a key feature of mRNA buffering. Our model proposes that many mRNAs are exported to the cytoplasm while already carrying some of the degradation factors with them. We propose that the Kis value reflects the propensity of nuclear Xrn1 and

its associated factors to bind the cognate promoters[2,3] and the resulting nascent transcripts (see Fig. 7), and hence the proportion of mRNA molecules that carry these decay factors with them to the cytoplasm. Thus, most mRNAs are degraded by both the non-Kis pathway (the classical mRNA decay pathway) and the Kis pathway. Taken together, our results suggests that nuclear factors, including the transcription apparatus, are capable of regulating cytoplasmic mRNA decay by controlling mRNA imprinting of Xrn1.

The Kis pathway represents a forward pathway connecting transcription and mRNA decay. We also uncovered a component of an inverse pathway connecting mRNA degradation with transcription, whereby binding of RNA to NLS1 appears to regulate Xrn1 import. More specifically, we provide evidence that RNA (containing a 5′phosphate), which binds both the active site and the R3H-like motif in a cooperative manner (Fig. 3c, compare lanes "R101G" and "S454P", with the lane "R101G, S454P"), also binds NLS1 (Fig. 3b, see lane "ΔNLS1") and inhibits the Kap120-Xrn1 interaction (Fig. 4f). This establishes a linkage between mRNA decay and a retrograde pathway involving Xrn1 import and subsequent transcription. As the two Xrn1 NLSs are required for import, we propose that when RNA is covering NLS1, import is repressed; only following RNA degradation NLS1 is exposed and, if also NLS2 is exposed, Xrn1 is imported to the nucleus. We cannot rule out the possibility that also Kap120-NLS2 interaction is regulated, raising an option of more complex regulation of Xrn1 import. Interestingly, blocking Xrn1 import, which is likely to increase Xrn1 concentration in the cytoplasm, did not contribute much to degradation of non-Kis mRNAs because only a minor population of mRNAs (-100) become >2-fold unstable in the NLS1/2 mutant (see Supplementary Data 4). Thus, it seems that the cytoplasmic concentration of Xrn1 is not limiting for the classical decay pathway. Relevant to our results is the observation that, in human cells, massive mRNA degradation due to viral infection can lead to enhanced import of poly(A)-binding proteins[38,39]. One of them, PABPC, is imported by importin α whose interaction with PABPC NLS is compromised by the poly(A) tail[40]. Perhaps, by virtue of the capacity of the negatively charged RNA to electrostatically interact with positively charged classical NLSs, RNA can control import of various shuttling RNA-binding proteins. Nevertheless, it appears that RNA-classical NLS interaction is weak and, at least for NLS1, requires a nearby RNA-binding motif (R3H-like) for stabilization. Indeed, Xrn1^ΔNLS1/2 seems to bind mRNA normally in vivo, consistent with a small contribution of NLS1 to the overall Xrn1 binding capacity.

During constant environmental conditions, Xrn1 shuttling plays little role in mRNA buffering and has no effect on proliferation rates. However, Xrn1 shuttling plays a role during transitions from one condition to another - illustrated here by two examples. The first is a transition from starvation (stationary phase) to sated conditions where *xrn1*^ΔNLS1/2 cells exit more slowly and abnormally from starvation (Fig. 6a-d). The second example is fluctuating temperatures (Fig. 6i). Dynamic environmental alternations, such as changes in ambient temperature or nutrient availability, represent a fundamental challenge to living organisms, including yeast. Under ever-changing conditions, that better represent the real yeast habitat, the proliferation rate of *xrn1*^ΔNLS1/2 cells is abnormally slow. In contrast, the mutant cells can proliferate like wild-type at any temperature that we tested, provided that it does not change (Supplementary Fig. 5 f-h). In response to various stresses, mRNA buffering transiently collapses, while the level of mRNAs transiently changes, resulting in a "peak behaviour"[41–45]. Peak behaviours can also be obtained by abrupt changes in the levels of Xrn1 and some of its associated factors[46]. This behaviour is important for properly coping with the stress (reference 48 and references therein). It was previously demonstrated mathematically that, given a constant TR, the kinetics of the peak behaviour in response to the environment is solely dependent on mRNA decay rate; a faster rate results in a faster response, i.e., the time it takes for mRNA levels to reach their highest

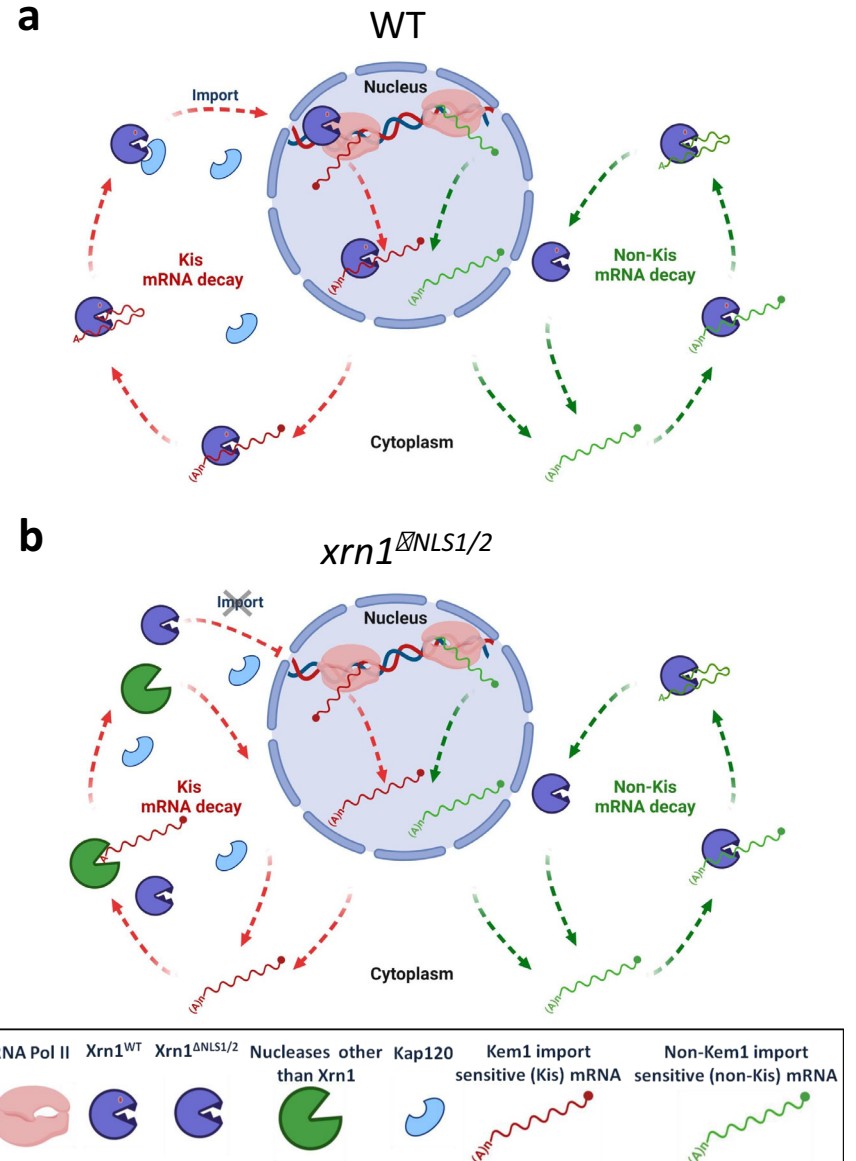

**Fig. 7 | A proposed model for two Xrn1-mediated mRNA decay pathways.**
**a** *WT cells.* Left pathway (red arrows) illustrates the Kis mRNA decay pathway, which begins in the nucleus. Xrn1 binds promoters and stimulates transcription[1–3]. During transcription, it binds the emerging transcript[73], most probably at its 3' end[27]. This RNA is protected against Xrn1 activity by a 5' cap, yet Xrn1 is immediately available upon decapping, or after endonucleolytic RNA cleavage. In this model, we hypothesize that Xrn1 does not dissociate from binding the 3' end. Thus, upon decapping, Xrn1 binds both the 5'P and the 3' RNA end (until advance stage of the RNA degradation). This hypothesis remains to be examined experimentally. Following

RNA decay, NLS1 is exposed and binds Kap120, and Xrn1 is imported to begin a new cycle. Right pathway (green arrows): the decay of non-Kis mRNAs is, by definition, insensitive to Xrn1 shuttling. It is confined to the cytoplasm and follows the standard model of mRNA decay[8,9]. **b** *xrn1ΔNLS1/2 cells.* Kis mRNAs are degraded mainly by the Xrn1-independent pathway, quite possibly by the exosome - as it becomes essential in *xrn1ΔNLS1/2* cells (Fig. 5k). In general, we propose that most mRNAs are simultaneously degraded by the two pathways, the impact of each pathway on the overall degradation is mRNA specific and is related to the "Kis value". See text for more details. Created with "BioRender.com".

value is directly affected by the decay rate[47,48]. In case both mRNA synthesis and decay rates increase, the kinetics of mRNA levels changes are even faster[41,47]. Therefore, it seems that the effect of Xrn1 import on enhancing both mRNA synthesis and decay in response to environmental changes is related to the kinetics of the response and not to the steady-state levels of these mRNAs, after the cells adapt to the new environment. This is in line with the proposal that higher turnover allows faster response to the environment regardless of the actual steady-state mRNA levels[47,49]. We therefore propose that a main function of Xrn1 shuttling, and its associated mRNA decay proteins, under changing conditions is to enable rapid acquisition of new steady-state levels. However, once the new levels are obtained, Xrn1 shuttling is neutral for subsequent proliferation. Yeast cells are

highly adaptive and can obtain homeostasis in a broad range of conditions if enough time is provided[46,50]. However, in their native environment, the rate of obtaining this homeostasis is likely important for survival and out-competing other yeasts. We hypothesize that regulated shuttling of molecules between the two major compartments has been evolved, in part, to enable rapid response to the environment and perseverance in the face of constant environmental uncertainty.

## Methods
### Yeast Strains and Plasmids construction
Yeast strains, and plasmids are listed in Supplementary Data 1 and 2. Xrn1 NLSs were mutated by homologous recombination with PCR amplified mutated fragments, primers carrying mutations were used

to incorporate the mutations. For creating yeast strains, Xrn1 (or, gene of interest) is replaced by *CaURA3*. WT or mutated fragments was later introduced by replacing *CaURA3* by homologous recombination and selection on 5-FOA. In this way, mutations were surgically introduced without adding other sequences. All strains were verified by both PCR and Sanger sequencing.

## Antibodies used in this study

Anti-6xHis Monoclonal Antibody (Takara, 631212; used at 1:5000 dilution) https://www.takarabio.com/products/protein-research/antibodies-and-immunoprecipitation/myc-ha-and-6xhis-tag-antibodies Anti-FLAG Monoclonal Antibody (Genscript, A00187; used at 1:1000 dilution), Clone ID: 5A8E5 (https://www.genscript.com/antibody/A00187_100-_THE_sup_TM_sup_DYKDDDDK_Tag_Antibody_mAb_Mouse.html). Anti-GFP Antibody (previously, Covance, MMS-118P; used at 1:5000 dilution) (Now, BioLegend, 902601), Clone: B34 https://www.biolegend.com/en-us/products/purified-anti-gfp-epitope-tag-antibody-11364 Goat Polyclonal Anti-Mouse IgG Peroxidase-conjugate (Sigma Aldrich Cat# A3682; RRID:AB_258100; used at 1:5000 dilution) (https://www.sigmaaldrich.com/IN/en/product/sigma/a3682) Goat Polyclonal Anti-Rabbit IgG HRP-conjugate (Sigma Aldrich Cat# 12348; RRID:AB_11214240; used at 1:10000 dilution) (https://www.sigmaaldrich.com/IN/en/product/mm/12348?gclid=CjwKCAjwyaWZBhBGEiwACslQo-rQz85vZ53QFqWIvn9_FnqM_2IpQ-bnjyUrVIIRr2fZ3netxOs2yxoCSvkQAvD_BwE&gclsrc=aw.ds) Anti-Pat1 antibody was kindly donated by Prof. Roland Lill, Germany (Rodriguez-Cousino N., et.al., Yeast, 1995; used at 1:1000 dilution) Anti-DHH1 antibody was kindly donated by Prof. Karsten Weis, Germany (Fischer N., et.al., Embo J, 2002; used at 1:5000 dilution).

## Yeast proliferation conditions, under normal and fluctuating temperatures, and during exit from starvation

Yeast cells were grown in synthetic complete (SC), Synthetic dropout (SD) or in YPD medium at 30 °C unless otherwise indicated; for harvesting optimally proliferating cells, strains were grown for at least seven generations in log phase before harvesting. For the nucleocytoplasmic shuttling assay, cells were grown at 24 °C before shifting to 37 °C for 2 h. For the nutrient starvation experiment, we employed previously described protocol;[3] briefly, exponentially growing cells were washed once with water and then resuspended in Synthetic media lacking both carbon source and amino acids. Cells were imaged after 1 h with a fluorescence microscope. For exit from long-term starvation, cells were proliferated in SD medium at 30 °C and allowed to enter stationary phase. Following entry, cells were incubated for 7 additional days. To re-feed the culture, cells were collected by centrifugation and were resuspended in fresh medium at 10 fold smaller cell density, followed by incubation at 30 °C. Yeast proliferation was measured by counting the buds with a phase-contrast microscope. For fluctuating temperature assay, cells were grown in YPD over-night till mid-log phase. Cells were then cultured in 100 µl of liquid-rich medium (YPD) in several PCR tubes, each tube was used for measuring one-time point, at initial $10^4$ cells/ml. After 10 min acclimation at 30 °C, the temperature cycled between 38 °C for 150 min and 8 °C for 150 min. At the indicated time points, a tube was taken and cell density was determined by counting cells using haemocytometer. $N > 200$ cells.

Synthetic sickness assay

Optimally proliferating cells that were taken from tiny colonies, developed during over-night incubation at 30 °C on an SC plate, were streaked on plates with or without auxin and incubated for 2 days at 30 °C prior to photography.

## Nucleocytoplasmic Shuttling Assay

Cells were allowed to proliferate at 24 °C until 3-5 × 10⁶ cells/ml, rapidly shifted to 37 °C and subsequently incubated for up to 2 h. The translation inhibitor cycloheximide (CHX) (Sigma) was added to a final concentration of 300 µM just prior to the temperature shift. Pab1-GFP, a shuttling protein[23] was used to determine the efficiency of the assay and to serve as a nuclear marker[3,23]. In all cases, Pab1-GFP was present in the nuclei of at least 60% of the heat-inactivated mutant cells after 2 hr. To determine whether nuclear proteins shuttle, we used nup49-313 cells; when shifted to 37 °C these cells cannot import proteins from the cytoplasm to the nucleus[26].

## Identifying Tail mutations that led to constitutive nuclear localization

A PCR-based random mutagenesis protocol[51–53] was used to mutate Tail-GFP fragments. The mutants were introduced into a WT strain (yMC229) and analysis was performed by fluorescence microscopy followed by sequencing of nuclear-localized isolates.

## Fluorescence microscopy

Fluorescence microscopy was performed as described before[3,54]. Image processing and quantitation when required was done by ImageJ[55,56]. The nuclear/whole-cell ratio of the fluorescent signal was determined by ImageJ. The nuclear and cellular boundary (ROI) was defined by hand-drawn tool of ImageJ, followed by measuring the intensity

## Affinity purification of Kap120-HTP, Xrn1-FLAG and in vitro interaction assay between the purified proteins

Cells were grown in YPD media until 2-2.5 × 10⁷ cells/ml and were harvested by filtration and flash frozen by plunging into liquid nitrogen. Frozen cells were lysed cryogenically via six cycles of pulverization (15 herz) using a mixer mill 400 (RETSCH). Grindates were later resuspended in appropriate lysis buffer supplemented with protease inhibitors, centrifuged (8000 g for 10 min) and protein amount was measured. Subsequently equal amounts of protein were subjected to IP. FLAG-IP was done using FLAG kit (Sigma, Cat.No: FLAGIPT1), according to manufacturer's protocol. Briefly, protein samples were rotated overnight at 4 °C with anti-FLAG M2 sepharose beads. Sepharose suspensions were centrifuged for 30 seconds at 1000 g, supernatant discarded and subsequently washed 4 times with 1X Wash Buffer. IP was eluted twice with 3X FLAG Elution Buffer and stored at -70 °C until further use. Protein-A tagged proteins were affinity purified by incubating with IgG sepharose resin for 2 h at 4 °C. The resins were washed 4 times with IPP150 (10 mM Tris-Hcl pH 8.0,100 mM Kac, 0.1% NP-40, 2 mM MgAc,10 mM β-Mercaptoethanol, 1 mM PMSF), followed by washing once with TEV cleavage buffer (10 mM Tris-HCl pH 8.0, 100 mM Kac, 0.1% NP-40, 1 mM MgAc,1 mM DTT) without the TEV. The elution was made with by incubating the beads in TEV buffer with TEV protease for 2 h at room temperature. Eluates were stored at -70 °C until further use. His-tagged protein purification was done with Ni-NTA agarose resin (Qiagen) according to manufacturer's protocol; briefly, overnight bacterial culture with Kap120-6xHis was grown for few hours. At 0.6 O.D., 0.1 mM IPTG was added. Four h later, cells were harvested and resuspend very gently with 10 ml ice cold 1x PBS, spinned 8000 rpm for 10 min at 4 °C. The pellet was frozen at -80 °C. To extract proteins, the frozen cell pellet was resuspended in ice cold 30 ml of lysis buffer (50 mM Tris-HCl, pH 8.0, 250 mM NaCl, 0.5% Triton-X-100,10 mM Imidazole + protease inhibitors). Cold cells were subjected to 6 pulses of sonication -10 sec each- with intervals of 30 sec in ice, spinned for 5 min at 13000 rpm 4 °C. Seven ml of unpacked Ni-NTA resin was washed with 2 × 7 ml lysis buffer and the supernatant was mixed with the resin for 1 h in a rotator at 4 °C. The mix was transferred to a column and beads were washed with 3 × 10 ml Wash-buffer (50 mM Tris-HCl, pH 8.0, 250 mM NaCl, 0.5% Triton-X-100, 20 mM Imidazole, pH was adjusted to 8.0 using NaOH). Protein was eluted twice with 1 ml elution buffer (50 mM Tris-HCl, pH 8.0, 250 mM NaCl, 0.5% Triton-X-100, 250 mM Imidazole, pH was adjusted to 8.0 using NaOH).

For in-vitro interaction assays, purified His-tagged Kap120 was mixed with Xrn1-FLAG (WT or mutant) protein and incubated in a rotator for 1 h at 4 °C. To pull down Xrn1-FLAG/Kap120-HISx6 complex, anti-FLAG coated beads or Ni-NTA agarose beads were used, and proteins were eluted with 2X LSB and heat. For experiments involving RNA, a commercially synthesized 40 nt long RNA (5'-AUGUACA-GUCCGACGAACACGGAGUCGACCCGCAACGCGA-3') was added to the mix, EDTA (2 mM final concentration) was added to prevent degradation of RNA by Xrn1. Similarly, GFP-tagged Tail was pulled down with GFP-Trap® resin (ChromoTek) in a small column, washed 3 times with 1X Wash buffer. Purified Kap120-His was added to the columns and incubated in a rotator for 1 h at 4 °C. The column was washed and eluted with 2X LSB at 95 °C.

## Western blotting analysis

Proteins were run in a precast SDS-PAGE gel (BioRad), followed by an electro-transfer to a Nitrocellulose membrane (BioTrace™ NT; PALL Life Sciences). The membrane was blocked for 1 h with non-fat milk at room temperature or over-night at 4 °C. It was then reacted with appropriate primary antibody O.N at 4 °C, or 1 h at room temperature, washed and subsequently reacted with HRP conjugated secondary antibody. Band intensity was detected with the Western Lightning Plus-ECL (Perkin Elmer) according to the manufacturer's instructions. The membrane is exposed to either X-ray film or photographed with ImageQuant™ LAS4000 (GE Healthcare).

## RNA immunoprecipitation – seq (RIP-seq) approaches

In vivo or in vitro mRNA interactome profiles of FLAG-tagged WT and ΔNLS1/2 mutant were determined, each condition in duplicate, as follows. Condition I: in vitro. Purified Xrn1 samples were reacted in vitro with affinity purified poly(A) + cellular mRNAs (purified using oligo(dT)$_{20}$ beads) followed by RIP-seq. Condition II: in vivo. *XRN1* WT or *XRN1*$^{ΔNLS1/2}$ mutant, genomically tagged with FLAG and 6His were UV crosslinked in-vivo using Vari-X-Link UV-crosslinker. IP was done in Tandem with anti-FLAG magnetic beads followed by His purification with Ni-NTA resin in denaturing condition. RNA was released by incubating the bead-conjugated RNA-protein complex with Proteinase K at a final concentration of 100 μg/ml for 1 h at 56 °C. Total RNA was purified with standard Phenol:Chloroform extraction and precipitated in presence of 30 μg of Glycoblue. To ascertain interaction of Xrn1 with poly(A)-tail containing RNAs, we created libraries utilizing (dT)$_{20}$ containing primer (oMC 2529: GTGACTGGAGTTCAGACGTGTGCT CTTCCGATCTTTTTTTTTTTTTTTTTTTTTTVN). Following sequencing, we selected reads that contained >15 adenines that represent mainly full length mRNAs, as degradation intermediate usually contain 10 or less adenines[8]. For each gene, the number of RIP-seq reads within 250 bp of the annotated primary polyadenylation sites was obtained in each replicate, and these values were added together.

## Identifying proteins that interact with the nuclear Xrn1

Strains expressing genomically FLAG-tagged WT or ΔNLS1/2 Xrn1, as well as no-tag control, were grown in duplicates in YPD for 7 generations till they reach mid-log phase and were harvested and grinded. Equal amount of protein lysate was aliquoted and subjected to immunoprecipitation, using Sepharose coupled anti-FLAG Abs (FLA-GIPT1, Sigma), according to the manufacturer's protocol After competitive elution with 3x FLAG peptide, elutes were run in a gel and respective lanes were cut into pieces and subjected to in-gel digestion, followed by mass spectrometric identification of the proteins. MS data were analyzed using MaxQuant program. Statistical analysis was done using Perseus software with the volcano plot option as follows: Label-free quant (LFQ) values of MS analyses were transformed to log2 scale. After filtering out proteins that did not appear in at least 1 group, missing values were imputed from normal distribution. The WT was compared to the control (no tag) using t-test while applying

permutation-FDR correction with FDR = 0.05 and S0 = 0.1. The NLS mutant was compared to the control (no tag) using t-test while applying permutation-FDR correction with FDR = 0.05 and S0 = 0.1. Proteins with log2 [fold change] > 1 and p value < 0.1 were considered significantly enriched. The WT was then compared to the NLS using t-test while applying permutation-FDR correction with FDR = 0.05 and S0 = 0.1.

## RNA-protein interaction assays

One liter of culture was harvested at $1 × 10^7$ cells/ml and ground cryogenically as described above. For UV crosslinking, the grindate was transferred to a glass petri-dish and mixed with powdered dry-ice, irradiated 3 times in an UV-crosslinker (Stratalinker) with 0.6 J/cm². During intervals, more dry-ice powder was added to replenish evaporated dry-ice, keeping the cell powder frozen during the crosslinking. The grindate was then suspended in lysis buffer (6 mM Na$_2$HPO$_4$, 4 mM NaH$_2$PO$_4$.H$_2$O, 1% NP-40, 100 mM Potassium Acetate, 2 mM Magnesium Acetate, 50 mM Sodium Fluoride, 0.1 mM Na$_3$VO$_4$, 10 mM β-mercaptoethanol, protease and phosphatases inhibitors, centrifuged at 20,000 g for 15 min at 4 °C and supernatant was collected. Equal amount of proteins were mixed with IgG epharose beads, incubated for 1 h at 4 °C. The samples were later transferred to the small empty columns (Bio-Rad) and washed 3 times with lysis buffer and once with TEV cleavage buffer. 1U of Rnase-cocktail (RnaseA + T1, Thermo) was added to the beads for 5 min at 37 °C, reaction was stopped by adding 650 mg guanidium-HCl (final concentration 6 M) to each of the reaction tube. NaCl and imidazole was added to a final concentration of 300 mM and 10 mM, respectively. Each sample was mixed with 100 μl Ni-NTA agarose beads and rotated O.N. at 4 °C. Beads were collected by centrifugation and washed twice with 500 μl wash buffer 1 (50 mM Tris-HCl pH 7.6, 300 mM NaCl, 10 mM imidazole, 6 mM Guanidinium-HCl, 0.1% NP-40, 5 mM β-mercaptoethanol) and 4 times with 500 μl PNK buffer (50 mM Tris-HCl pH 7.6, 10 mM MgCl$_2$, 0.5% NP-40, 10 mM β-mercaptoethanol). 80 μl of PNK buffer containing 4U Thermosensitive Alkaline Phosphatase (TSAP) (Promega) and 2 μl Rnasin® (Promega) was added to the beads and incubated for 60 min at 37 °C. Beads were washed once with 500 μl 1X Wash Buffer (same as above) and 3 times with 500 μl PNK buffer. Eighty μl of 1X PNK containing 2 μl (20U) of T4 PNK (New England Biolabs), 2 μl (80U) Rnasin and 20 μCi γ32P-ATP (PerkinElmer) were added to the beads pellet and incubated 60 min at 37 °C. Beads were washed once with 500 μl Wash 1 followed by 3 washes with 500 μl PNK buffer, and eluted with 100 μl elution buffer (50 mM Tris-HCl pH7.6, 50 mM NaCl, 200 mM imidazole, 0.1% NP-40, 5 mM β-mercaptoethanol) for 5 min at room temperature.

## Electrophoretic mobility shift assay (EMSA) (gel-shift)

Purified Xrn1 was diluted with binding buffer (50 mM Tris-HCl, pH 7.4, 150 mM NaCl, 2 mM DTT, 1 mM EDTA) in a total volume of in 10 μl. 5'-end labelled-probe[57] was heated at 95 °C for 1 min, and snap chill in wet ice for 5 min. 100 CPS of radiolabelled probe, in a binding buffer, was added to each reaction tubes. The mixture was incubated for 30 min at room-temperature. One μl of loading buffer (80% glycerol + bromophenol blue + Xylene-Cyanol) was added and samples were fractionated by 6% non-denaturing polyacrylamide gel in 0.5 x TBE buffer. The gel was run for 3 h at 200 V in 0.5 X TBE in the cold-room. The gel was exposed to X-ray film at -70 °C.

## Crosslinking and analysis of cDNA (CRAC)

The CRAC experiment and its analyses were performed as previously described[58–61]. Briefly, optimally growing cells carrying appropriate variations of HTP-tagged Xrn1 or Tail were UV crosslinked and harvested. RNA-protein complexes were captured tandemly on IgG sepharose and Ni-NTA followed by partial RNase digestion using RNace-IT. After ligation of sequencing adaptors, cDNA libraries were prepared by reverse transcription and PCR amplification followed by Illumina deep

sequencing. Following the collection of *.fastq files, adapter trimming was done by flexbar[60,61]. RA3 (5'-TGGAATTCTCGGGTGCCAAGG-3') and RA5 (5'-GTTCAGAGTTCTACAGTCCGACGATCNNNNNAGC-3') adapter of Illumina TruSeq were selected as input for the flexbar run. For further sequence processing and statistical analysis, pyCRAC packages were used (https://git.ecdf.ed.ac.uk/sgrannem/pycrac). pyFastqDuplicateRemover.py, a program, which removes duplicate from the flexbar trimmed files were used for sequence processing (pyCRAC package). Next, Bowtie2 v2.4.2 was used to align trimmed sequence on R64-1-1 reference genome (https://uswest.ensembl.org/Saccharomyces_cerevisiae/Info/Annotation) and *.sam output format was produced for the downstream analyses. pyReadCounters.py program (from the same pyCRAC package) was used to calculate the RPKM values of genes.

### Genomic run-on

Genomic Run-On (GRO) was done in three biological replicates essentially as decribed[62,63]. Exponentially growing yeast cells (total of $5 \times 10^8$ cells at $OD_{600} = 0.5$) were taken for each run-on reaction. Another aliquot of the same cells was used directly for RNA extraction, which subsequently was used for cDNA synthesis using oligo-dT as the primer and $^{33}$P-dCTP to determine mRNA abundance (RA). GRO samples provided nascent transcription rates (nTR) for every yeast gene. Values were normalized to the average cell volume (the median of the population measured by a Coulter Counter device), although this correction had very little effect since cell volume of all the studied strains was similar. Whole RNA polymerase II TR was obtained by summing up all individual genes TR data. Ras were obtained from the hybridization of labelled cDNA onto nylon filters. Total mRNA concentration in yeast cells was determined by quantifying polyA + in total RNA samples by oligo-dT hybridization of a dot-blot following the protocol described[64] and dividing by average cell volume. mRNA half-lives (HLs), in arbitrary units, were obtained by dividing individual RA values by TR ones.

### mRNA decay assay

mRNA decay assay was performed as described previously[10,17,65]. To block transcription of non-stress genes, optimally proliferating cells (-1 × 10^7 cells/ml) were shifted rapidly from 30 °C to 42 °C by shaking at 70 °C until the culture reaches 40 °C and then incubated at 42 °C. Samples were taken at various time points post-temperature shift and mRNA levels were determined by Northern blot hybridization and quantified by PhosphoImager technology as described previously[54].

### Cell-cycle analysis

Cells were proliferated in SD medium at 30 °C and allowed to enter stationary phase (SP). Following entry, cells were incubated in for 7 additional days. Cells were collected by centrifugation and were resuspended in fresh medium at 10 fold smaller cell density, followed by incubation at 30 °C. 25 ml of Cells were withdrawn at each timepoint and concentrated 3-fold. Cells were mildly sonicated to break the clumps. 300 μl of cells were transferred to microcentrifuge tube, centrifuged and resuspended in 700 μl ethanol (final 70% ethanol), incubated at room temperature for 2 h. Fixed cells were centrifuged, and the pellet was washed 4 times with 1 ml of 0.2 M Tris HCl (pH 7.5). Pellet was resuspended in 0.5 ml Tris buffer. 10 μl 0.5 M EDTA and 50 μl RnaseA (10 mg/ml), and rotated overnight at 37 °C. Cells were centrifuged and resuspended in pepsin (5 mg/ml in HCl) and incubated for 2 h at 37 °C. Cells were washed with 0.2 M Tris and eventually were suspended in 0.3 ml of 0.2 M Tris. Cells were diluted 1:10 in 1 ml 0.2 M Tris pH 7.5, 1 μM SYTOX[66–68] was added just before injection into the flow cytometer (Beckton-Dickinson FACScalibur; BD FACS Diva (version 8.0.3) FCS Express (version 7) ImageJ 1.49 v TotalLab Quant 2.2). 10,000 ungated events for each run were

recorded using FL1 (488 nm laser). The data were analysed by FCS Express (DeNovo software). See "Reporting Summary" in Supplemental information for more details.

### Determining mRNA level during exit from starvation

Cells starved for 7 d were re-fed as described above, in duplicates. Cell samples were taken at 0, 0.75 and 2.25 h post-refeeding. *S. pombe* cells were spiked-in (10% of the cells). mRNAs were extracted and purified by oligo(dT) and sequenced (see "RIP-seq approaches" above). mRNA levels were normalized to the spike-in.

### Data analysis and statistics

Fluorescence images were analyzed by ImageJ[55,56].' Immunoblotting and northern-blotting band intensities were quantitated by TotalLab software. SwissModel[69] was used to model Xrn1 and Tail, PyMol (The PyMOL Molecular Graphics System, Version 1.2r3pre, Schrödinger, LLC) was used to visualize and manipulate the model and for overlaying of structures. To determine sequence conservation, sequence logo was generated, using the online WebLogo program[70] (http://weblogo.berkeley.edu/logo.cgi). Protein disorder was predicted in PONDR[71,72] (http://www.pondr.com/). Data was plotted and statistical calculation were done in Excel or R environment using 'ggplot2', 'eulerr', 'ggpubr' and 'plyr' packages. For assessing the difference between two groups, *p*-value was calculated using Student's unpaired T-test or Wilcoxon rank sum test (Also known as Mann-Whitney U test) as indicated in the Figure legends.

## Data availability

The data that support this study are available from the corresponding author upon reasonable request. RNA sequencing data from CRAC experiment was uploaded in the NCBI-SRA repository, BioProject ID: PRJNA875575 RIP-seq and RNA-seq data was uploaded in the ArrayExpress repository with accession number E-MTAB-12110 for the RIP data and E-MTAB-12111 for the total RNA starvation experiment data. Raw data for GRO experiment was uploaded in the GEO repository, accession number for NLS mutant experiment is GSE158250, GSE204991 is for delta kap120 experiment. Proteomics Raw data has been submitted to PRIDE database with accession number PXD036414. Source data are provided with this paper.

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

## Acknowledgements

We thank Juana Diez and Orna Amster-Choder for critically reading the manuscript. We thank Philipp Hackert for help with the CRAC experiments. This work was supported by the Israel Science Foundation (1472/15) to MC and by the Deutsche Forschungsgemeinschaft (SFB1190 to MTB), the Heidenreich von Siebold program of the University Medical Center Göttingen (to KEB) and grant PID2020-112853GB-C31 funded by MCIN/AEI/10.13039/501100011033 to J.E.P.-O. It was supported in part at the Technion by a fellowship of the Israel Councel for Higher Education (to SC).

## Author contributions

S.C. and M.C. conceived the study; M.C. supervised the study; J.G.M. and J.E.P. designed, performed and analyzed the various GRO experiments; G.H. identified NLS1 and performed initial shuttling experiments; J.F. analyzed the in vivo/in vitro Xrn1-RNA binding data; A.K. help identifying Kap120 as Xrn1 importin and performed the Mass spec experiment; O.B. performed screening of point mutants, and also performed RNA stability experiment; S.G.C. and M.S. performed automatic screening of Tail-GFP localization; R.E. analyzed the CRAC data; M.I.R. identified the R3H-like motif; S.U. analyzed the Mass spec data; S.D. perfomed some of the fluorescence microscopy experiments and help in preparing NGS libraries; K.E.B. and M.T.B. performed the CRAC experiment. S.C. and M.C. designed all other experiments while S.C. performed them and analyzed the data. S.C. and M.C. wrote the paper, and all authors edited and/or approved the manuscript.

## Competing interests

All authors declare no competing interest.
