## [Peer Review File · Nature Communications]

Reviewers' comments:

Reviewer #1 (Remarks to the Author):

Review for the manuscript NCOMMS-21-03251-T by Dr. Choder and co-authors entitled "RNA-controlled nucleocytoplasmic shuttling of mRNA decay factors regulates mRNA synthesis and initiates a novel mRNA decay pathway".

The regulation of mRNA levels is a critical feature of gene expression in all kingdoms of organisms. In recent years, "mRNA buffering", which maintain proper concentrations of mRNAs, have discovered reciprocal adjustments between the overall rates of mRNA synthesis and degradation. In the budding yeast *Saccharomyces cerevisiae*, a number of factors, known to degrade mRNA in the cytoplasm shuttle between the nucleus and the cytoplasm. The mRNA buffering mechanism is not restricted to factors recognized as mRNA decay factors, but it also includes components of the transcription apparatus. The authors demonstrated that Pol II regulates mRNA translation and decay by mediating Rpb4/7 co-transcriptional binding to Pol II transcripts, a process they named "mRNA imprinting". The components of the Ccr4-Not complex also imprint mRNA and regulate mRNA export, translation and decay. The basic idea is that cellular localization of these factors is critical for their two opposing activities, either in synthesis or degradation.

In this study, they demonstrated that Xrn1 nuclear import, mediated by two NLSs and by the decaying RNA, affects mRNA synthesis and decay of a large portion of the mRNAs named as Kem1 import-sensitive (Kis) mRNAs. Half-lives of those mRNAs are affected by the blocking of Xrn1 import, indicating that the nuclear function of Xrn1 is indispensable for Xrn1-mediated cytoplasmic decay. They propose that the decay pathway of Kis mRNAs begins in the nucleus, probably by Xrn1 binding to these mRNAs, suggesting that many mRNAs are exported to the cytoplasm while already carrying some of the degradation factors with them.

The proposal is potentially interesting. However, the quality of results is not sufficient enough to support their proposal that the nucleocytoplasmic shuttling mechanism of Xrn1 regulates mRNA synthesis and decay. There is no direct evidence to demonstrate that Xrn1 recognizes specific mRNA and determines/affects its stability or localization. It is also unclear how nucleocytoplasmic shuttling is regulated by Kap120. There is no clear mechanistic connection between the blocking of Xrn1 import and inefficient synthesis and decay of mRNAs. Finally, the blocking Xrn1 import did not affect mRNA levels. Therefore, the physiological significance of the regulation is unclear. More experiments are needed to be a candidate for publication in Nature Communications.

Comments:

1. The physiological significance of the nucleocytoplasmic shuttling of Xrn1:

The authors proposed that the blocking Xrn1 import leads to inefficient synthesis and decay of a large portion of the mRNAs without affecting mRNA levels (Fig. 5). The precise and comprehensive comparison between Xrn1-binding, synthesis, and decay of mRNAs are informative and crucial to evaluate the proposed model that cytoplasmic mRNA decay begins in the nucleus. The determination of protein levels derived from Kem1 import-sensitive (Kis) and non-Kis mRNAs by SLIAC may uncover the crucial function of the nucleocytoplasmic shuttling of Xrn1.

2. The physiological significance of the nucleocytoplasmic shuttling of Xrn1 in proliferation:

Xrn1 Δ NLS1/2 cells exit starvation slowly and abnormally upon re-feeding and are defective in proliferation at fluctuating temperatures (Fig.6). To understand how the nucleocytoplasmic shuttling of Xrn1 plays crucial roles in proliferation, the authors should determine the synthesis and decay and levels of Kem1 import-sensitive (Kis) and non-Kis mRNAs. Since it is possible that the blocking Xrn1 import leads to inefficient synthesis and decay without affecting mRNA levels in proliferation, the determination of protein levels derived from Kem1 import-sensitive (Kis) and non-Kis mRNAs by SLIAC may be required.

Minor points:

1. Nuclear localization of decay factors coupled with the nucleocytoplasmic shuttling of Xrn1: Nuclear localization of Xrn1 is eliminated by Δ NLS1 or Δ NLS2 Xrn1 cells (Fig. 1). The nuclear/whole-cell ratio of Pat1 and Dcp1, Lsm1 in Δ NLS1 or Δ NLS2 Xrn1 cells should be demonstrated to demonstrate nuclear localization of Xrn1 is crucial for the shuttling of decay factors. In addition, Tail-M457T-GFP is localized constitutively in the nucleus (Fig. 2.). The nuclear/whole-cell ratio of decay factors in Xrn1-M457T cells should be demonstrated.

2. Factors responsible for nucleocytoplasmic shuttling of Xrn1:

Δ NLS2-Xrn1 binds to Kap120 but is defective in nuclear localization (Fig.4). The authors must clarify the factor responsible for the nuclear localization of Δ NLS2-Xrn1. Tail-M457T is constitutively localized in the nucleus, and it is better to identify the factor responsible for the M457-dependent nuclear export of Xrn1.

Reviewer #2 (Remarks to the Author):

Chattopadhyay et al. examine the functional role of Xrn1 subcellular localization in linking mRNA transcription and decay and come to the conclusion that Xrn1 import is critical for cytoplasmic decay of a subset of cellular mRNAs ("Kis mRNAs"). They propose that in the context of sudden environmental fluctuations, Xrn1 shuttling is necessary to mediate proper mRNA transcription and decay dynamics. The authors convincingly identify and characterize 2 NLS in Xrn1 and show that shuttling of Xrn1 is likely responsible for bringing other mRNA decay factors into the nucleus. They also demonstrate that the 'tail' NLS of Xrn1 binds RNA in concert with its active site and that mutation of the NLS domains alters transcription rates and half-lives of a subset of mRNA. However, the apparent overlap of the RNA binding domain and NLS complicates the separation of the RNA binding and shuttling activities and thus the interpretation of phenotypes ascribed to NLS domain mutants. Additionally, several experiments would benefit from additional controls to bolster their conclusions.

Major Points

1. The authors' main premise is that Xrn1 import is needed to mediate Kis mRNA regulation; however, from these data it is unclear whether Kis mRNA dysregulation in $xrn1\Delta$ NLS1 cells is due to import defects or rather impaired binding of Xrn1 Δ NLS1 to Kis mRNAs. The latter explanation is supported by Figure 3b and 3d, which suggest Tail and presumably NLS1 is a determinant of Xrn1 binding specificity. Therefore, the authors must find some strategy to establish the importance of Xrn1 shuttling without disrupting the NLS and/or demonstrate that NLS1 deletion does not differentially affect binding to different sets of RNA, especially Kis and non-Kis RNAs. For example, they could perform the RNA half-life analysis in a Kap120 mutant with wildtype Xrn1, and/or analyze Tail-GFP's binding profile to see whether Kis RNAs are enriched among the bound RNAs (especially RNAs with high RPKM values). Additionally, they should indicate whether the binding profile is highly correlated with total RNA abundance in yeast cells, which may suggest Tail binds RNA non-specifically.

2. The shuttling assays with the Nup49-313 mutant (Figure 2g) would be significantly bolstered by including a positive control for nuclear import using other shuttling proteins (e.g. Pab1) to show that the conditions used indeed cause an import defect. Also important is inclusion of a control to demonstrate that these conditions affect only import, since Figure 2h shows significantly greater Tail nuclear retention at the restrictive temperature, suggesting export may also be impacted.

3. The data presented in Figure 4f do not strongly support the conclusion that RNA can outcompete Kap120 binding. The blot for Xrn1 seems overexposed, and the loading amount of Xrn1 Δ NLS2 seems to decrease progressively, mirroring the change in Kap120. This should be addressed by decreasing the image exposure to confirm consistent loading and additionally denoting the statistical significance in the corresponding bar graph. The loading variation issue also applies to Figure S3C.

Minor Points

1. Further clarification/description of the logic leading to each set of experiments in the results section would help increase clarity.
2. In general, the authors should clarify the number of cells counted for experiments (Fig1c-d) and include loading controls for Western blots from cell lysates.
3. Figure 2f should include TailM457TΔNLS1 mutant since M457T is the point mutant used for the subsequent shuttling experiments in Figure 2g. Alternatively, Figure 2g could include the S454P mutant.
4. Since graphs show discrete data points, they should be connected linearly, without smoothing.
5. RNA binding to an NLS as a mechanism to regulate the shuttling of mRNA-associated proteins has been previously reported (e.g. PABPC shuttling in mammalian cells), and the discussion may benefit from drawing parallels to these examples.

Reviewer #3 (Remarks to the Author):

Here the authors report that sequence analyses of Xrn1 predict 2 NLS motifs, which are shown to be functional in GFP fusion constructs. NLS1 is located in an R3H-like motif, which is shown to contribute substantially to RNA-binding. This interesting finding suggests competition between karyopherin and RNA binding, as previously reported for ribosomal proteins (DOI: 10.1093/emboj/21.3.377). Consistent with the reduced RNA binding, mutation of the NLS stabilized a subset of mRNAs. Surprisingly, the authors attribute impaired cytoplasmic nuclease activity to loss of nuclear import, rather than impaired substrate binding, which seems the obvious potential explanation. The authors then interpret all subsequent analyses and the conclusion in terms of the model that nuclear import is a prerequisite for Xrn1 functions. This interpretation would be consistent with the data, but is not demonstrated to be correct.

The findings on karyopherin binding - and the potential competition with RNA binding and Xrn1 function are of interest and significance. I would be more enthusiastic about publication of a shorter paper that focused on these findings.

Major point:

1: It was previously reported that "Rat1p and Xrn1p are functionally interchangeable exoribonucleases that are restricted to and required in the nucleus and cytoplasm, respectively" and that "targeting Xrn1p to the nucleus by the addition of the simian virus 40 large-T-antigen NLS resulted in complementation of the temperature sensitivity of a rat1-1 strain". (Johnston Mol Cell Biol, 1997). The conclusion of the previous work was clearly that Xrn1 is functionally restricted to the cytoplasm. Were these findings simply incorrect? It would be useful for the authors to refute the earlier findings, since the conclusions are so directly opposed.

2: An obvious potential interpretation of the effects of mutations in Xrn1 on mRNA stability is that they block interactions with the substrate and/or other components of the cytoplasmic mRNA degradation machinery. If the authors want to conclude that this is not the case, they need to provide some direct support. In the model that nuclear pre-mRNAs are stoichiometrically loaded with Xrn1, a block on import should cause very rapid loss of 5' mRNA turnover, but not 3' degradation.

3: The long, descriptive section on analyses of the cell cycle and growth conditions seems to be poorly

connected to the first sections of the paper and adds little. In all of these analyses, the contributions of reduced substrate binding vs reduced karyopherin binding, and direct vs indirect effects of large-scale changes in mRNA metabolism, of would need to be clearly distinguished for conclusions to be drawn. This would be better followed up in a separate paper.

4: It would be important to determine whether the proposed competition between important and RNA bunding actually takes place. Does karyopherin pre-binding inhibit RNA binding and degradation by Xrn1? Does RNA pre-binding block the karyopherin? These would be significant additions to the paper.

Minor point:

5: Page numbers would be helpful.

Responses to the Reviewers' comments

We would like to thank all the reviewers for their excellent suggestions that resulted in 22 new experiments and analyses. The revised manuscript includes the following new figure panels: Figs. 1e, 1f, 1g, 2g, 2h, 4g, 5i, 5j, 6g, 6h, S1f, S1g, S1h, S2e, S2f, S3d, S3e, S3f, S3g, S4i, S4m and S5e.

Reviewer #1 (Remarks to the Author):

I would like to thank Reviewer 1 for carefully reading our manuscript, for viewing our work as potentially interesting and for the constructive comments and suggestions. Our responses to Reviewer 1 comments are indicated below (the reviewer's comments are in black characters, whereas our responses – in blue).

Review for the manuscript NCOMMS-21-03251-T by Dr. Choder and co-authors entitled "RNA-controlled nucleocytoplasmic shuttling of mRNA decay factors regulates mRNA synthesis and initiates a novel mRNA decay pathway".

The regulation of mRNA levels is a critical feature of gene expression in all kingdoms of organisms. In recent years, "mRNA buffering", which maintain proper concentrations of mRNAs, have discovered reciprocal adjustments between the overall rates of mRNA synthesis and degradation. In the budding yeast *Saccharomyces cerevisiae*, a number of factors, known to degrade mRNA in the cytoplasm shuttle between the nucleus and the cytoplasm. The mRNA buffering mechanism is not restricted to factors recognized as mRNA decay factors, but it also includes components of the transcription apparatus. The authors demonstrated that Pol II regulates mRNA translation and decay by mediating Rpb4/7 co-transcriptional binding to Pol II transcripts, a process they named "mRNA imprinting". The components of the Ccr4-Not complex also imprint mRNA and regulate mRNA export, translation and decay. The basic idea is that cellular localization of these factors is critical for their two opposing activities, either in synthesis or degradation.

In this study, they demonstrated that Xrn1 nuclear import, mediated by two NLSs and by the decaying RNA, affects mRNA synthesis and decay of a large portion of the mRNAs named as Kem1 import-sensitive (Kis) mRNAs. Half-lives of those mRNAs are affected by the blocking of Xrn1 import, indicating that the nuclear function of Xrn1 is indispensable for Xrn1-mediated cytoplasmic decay. They propose that the decay pathway of Kis mRNAs begins in the nucleus, probably by Xrn1 binding to these mRNAs, suggesting that many mRNAs are exported to the cytoplasm while already carrying some of the degradation factors with them. The proposal is potentially interesting. However, the quality of results is not sufficient enough to support their proposal that the nucleocytoplasmic shuttling mechanism of

Xrn1 regulates mRNA synthesis and decay. There is no direct evidence to demonstrate that Xrn1 recognizes specific mRNA and determines/affects its stability or localization. It is also unclear how nucleocytoplasmic shuttling is regulated by Kap120. There is no clear mechanistic connection between the blocking of Xrn1 import and inefficient synthesis and decay of mRNAs. Finally, the blocking Xrn1 import did not affect mRNA levels. Therefore, the physiological significance of the regulation is unclear. More experiments are needed to be a candidate for publication in Nature Communications.

Comments:

1. The physiological significance of the nucleocytoplasmic shuttling of Xrn1: The authors proposed that the blocking Xrn1 import leads to inefficient synthesis and decay of a large portion of the mRNAs without affecting mRNA levels (Fig. 5). The precise and comprehensive comparison between Xrn1-binding, synthesis, and decay of mRNAs are informative and crucial to evaluate the proposed model that cytoplasmic mRNA decay begins in the nucleus. The determination of protein levels derived from Kem1 import-sensitive (Kis) and non-Kis mRNAs by SLIAC may uncover the crucial function of the nucleocytoplasmic shuttling of Xrn1.

The mechanism underlying Xrn1 activity in transcription initiation and elongation (including regulation of Pol II backtracking) was described in previous publications (Haimovich et al., *Cell* 2013; Fischer et al., *JBC* 2019; Begley et al., *RNA Biol* 2020). The function of Xrn1 shuttling on the linkage between mRNA synthesis and decay was not described in the literature and its effect on decay was surprising. In our original submission to this journal, we had performed mRNA synthesis and decay analyses, but, as Reviewer 1 indicated, we did not analyze the impact of blocking Xrn1 import on its RNA-binding capacity.

To obtain this missing link, we have determined Xrn1-binding capacity by RNA immunoprecipitation-seq (RIP-seq). To determine the possible linkage between this binding and the effect on mRNA synthesis and decay, we first assigned each gene with a "Kis value" that represents the sum of half-life (HL) ratio ($\Delta\text{NLS1/2 HL}/\text{WT HL}$) and inverse TR ratios ($\text{WT TR}/\Delta\text{NLS1/2 TR}$) (see lines 301-306). Thus, a Kis value represents the overall impact of Xrn1 import on mRNA synthesis and decay. [Just for clarification: the binary distinction between Kis and non-Kis is still valid; the Kis values simply represent a zoom in resolution]. Using the RIP-seq data and calculating the fold change of binding (mutant/WT), we found a correlation with the Kis values (Fig. 5j, "in vivo"). This was in contrast to binding of Xrn1 to purified mRNA in vitro (Fig. 5j, "in vitro"), indicating that this correlation is not an inherent feature of Xrn1, but rather it is a feature of Xrn1 cellular localization. This established a link between Xrn1 import and its RNA binding capacity that Reviewer 1 encouraged us to examine. In other words, Xrn1- mRNA binding capacity is affected by the Kis value which is related to the impact of blocking import on mRNA synthesis and decay.

To verify that Xrn1 shuttling activity is responsible for mRNA synthesis and decay rates, we performed a genomic run-on (GRO) analysis of cells carrying WT Xrn1 and lacking *KAPI20* (an experiment recommended by Reviewer 2). Consistent with the results of the NLSs disruption, also in this strain we observed defects in mRNA synthesis and decay (Figs. 4g and S3e-g), the extent of which is similar across the two types of experiments (compare Fig. 5a-b with S3e and g). These results emphasize the importance of Xrn1 shuttling on mRNA synthesis and decay in the context of WT Xrn1. As this strain expresses WT Xrn1, the effect on mRNA decay cannot be attributed to defects of Xrn1 in mRNA binding features.

We concur with Reviewer 1 that studying protein synthesis and degradation is important to understand the full system. However, since under optimal conditions mRNA levels as well as the proliferation rates are not affected by changing Xrn1 shuttling, we expect to see no difference between translation in WT and Xrn1^{ΔNLS1/2} cells or different translation rates between translation of Kis and non-Kis mRNAs. Moreover, if we find such differences, they would be difficult to interpret, as Xrn1 is involved in translation regulation in addition to its role in mRNA buffering (doi: 10.1038/s41467-019-09199-6).

The physiological significance of the nucleocytoplasmic shuttling of Xrn1 becomes apparent when cells need to respond to environmental changes, see our response to comment 2.

2. The physiological significance of the nucleocytoplasmic shuttling of Xrn1 in proliferation: Xrn1^{ΔNLS1/2} cells exit starvation slowly and abnormally upon re-feeding and are defective in proliferation at fluctuating temperatures (Fig.6). To understand how the nucleocytoplasmic shuttling of Xrn1 plays crucial roles in proliferation, the authors should determine the synthesis and decay and levels of Kem1 import-sensitive (Kis) and non-Kis mRNAs. Since it is possible that the blocking Xrn1 import leads to inefficient synthesis and decay without affecting mRNA levels in proliferation, the determination of protein levels derived from Kem1 import-sensitive (Kis) and non-Kis mRNAs by SLIAC may be required.

To study the possible role of Xrn1 nucleocytoplasmic shuttling on cell proliferation, we determined whether Xrn1 import affected the kinetics of mRNA accumulation shortly after starved cells were re-fed, i.e., when they just enter proliferation. We found that mutating NLS1/2 affected accumulation of a subset of mRNAs (Fig. 6g-h and S5e, Table S5). Thus, although, during optimal proliferation, mRNA synthesis and decay rates balance each other, during the transition from starvation to sated conditions there is no such a balance for a subset of genes and Xrn1 shuttling is important for a rapid accumulation of their transcripts.

The slow mRNA accumulation in XRN1^{ΔNLS1/2} cells can explain the slow exit of these mutant cells from starvation (Fig. 6a-c and S5a). Yet, once cells acclimate to the new environment, shuttling is dispensable for maintaining proper mRNA levels and growth. Provoked by these results, we then let cells proliferate under fluctuating temperatures and, again, observed slower proliferation of the mutant (Fig. 6i), despite normal proliferation at each of the extreme temperature provided that it does not change (!). In light of the results of these two experiments, we proposed that Xrn1 shuttling regulates gene expression in response to environmental changes (which often occur in the wild). This can provide one physiological relevance of Xrn1 shuttling.

As indicated above, SILAC is now not needed and will, anyhow, be difficult to interpret because Xrn1 is involved in translation (op. cit).

Minor points:

1. Nuclear localization of decay factors coupled with the nucleocytoplasmic shuttling of Xrn1: Nuclear localization of Xrn1 is eliminated by Δ NLS1 or Δ NLS2 Xrn1 cells (Fig. 1). The nuclear/whole-cell ratio of Pat1 and Dcp1, Lsm1 in Δ NLS1 or Δ NLS2

Xrn1 cells should be demonstrated to demonstrate nuclear localization of Xrn1 is crucial for the shutting of decay factors. In addition, Tail-M457T-GFP is localized constitutively in the nucleus (Fig. 2.). The nuclear/whole-cell ratio of decay factors in Xrn1-M457T cells should be demonstrated.

We have determined nuclear/whole-cell ratio of Pat1 Dcp1 and Lsm1 and found that disruption of either of the two NLSs affected their import, as for Xrn1 (Fig. 1e-g). These results are consistent with our conclusion that the two NLSs are required for Xrn1 import (disruption of one compromises import – Fig. 1c) and other decay factors (Fig. 1e-g). Consistently, the nuclear/whole-cell ratio of Xrn1^{M457T} and the other decay factors favors the cytoplasm (unlike localization of the small isolated tail), because only one functional NLS (NLS1) is available. See our responses to Minor point 2 below. We therefore expect that other mRNA decay factors will be localized in the cytoplasm as well.

2. Factors responsible for nucleocytoplasmic shuttling of Xrn1:

Δ NLS2-Xrn1 binds to Kap120 but is defective in nuclear localization (Fig.4). The authors must clarify the factor responsible for the nuclear localization of Δ NLS2-Xrn1. Tail-M457T is constitutively localized in the nucleus, and it is better to identify the factor responsible for the M457-dependent nuclear export of Xrn1.

The observations that rationalize the cytoplasmic localization of Δ NLS2-Xrn1 are: First, both NLSs bind Kap120 (Fig. 4). Second, Kap120 is required for efficient import of WT Xrn1 (Fig. 6f) and the tail domain alone (outside Xrn1 context) (Fig. 4e). Third, disruption of any NLS compromises Xrn1 import (Fig. 1C). Taken together, we propose that Kap120 is responsible for importing WT Xrn1 via binding of the two NLSs in a cooperative manner. Thus, both NLSs are required for import. Xrn1 ^{Δ NLS2} can bind only one Kap120 protein and therefore is cytoplasmic.

Here is our rationale for the need of two Kap120 proteins to import Xrn1: Xrn1 is a relatively large protein (~1450 AA). Large proteins often require >1 NLS to traverse the nuclear pore complex (reviewed in <http://dx.doi.org/10.1016/j.tibs.2015.11.001>). In contrast, Tail-GFP is a relatively small protein containing only NLS1. One NLS is sufficient to permit entry of a small protein (op. cit.). Thus, for Tail-GFP, NLS1 is required and sufficient; however, one Kap120 is not sufficient for importing the full length Xrn1. This is probably the reason why Xrn1^{M457T} is cytoplasmic at steady state, as only one functional NLS (NLS1) is present, which is not sufficient for import.

In response to the second part of this comment, we tried finding possible exportin(s) by affinity purifying Xrn1 and Xrn1 ^{Δ NLS1/2}. We hoped to find known exportin(s) that bind(s) only WT Xrn1 while in the nucleus. While we could find a number of interesting proteins that bind the WT but not the mutant Xrn1, including lowly expressed proteins (see Table S6), we could not find any exportin that bound WT Xrn1 but not Xrn1 ^{Δ NLS1/2}. In response to this comment, shuttling features are now better clarified in the Discussion chapter entitled: “Under optimal conditions, NLS1 and NLS2 seem to function cooperatively”.

Reviewer 2

I would like to thank Reviewer 2 for carefully reading our manuscript, and for the excellent and constructive comments and suggestions.

Our responses to Reviewer 2's comments are indicated below (the reviewer's comments are in black characters, whereas our responses – in blue).

Chattopadhyay et al. examine the functional role of Xrn1 subcellular localization in linking mRNA transcription and decay and come to the conclusion that Xrn1 import is critical for cytoplasmic decay of a subset of cellular mRNAs (“Kis mRNAs”). They propose that in the context of sudden environmental fluctuations, Xrn1 shuttling is necessary to mediate proper mRNA transcription and decay dynamics. The authors convincingly identify and characterize 2 NLS in Xrn1 and show that shuttling of Xrn1 is likely responsible for bringing other mRNA decay factors into the nucleus. They also demonstrate that the ‘tail’ NLS of Xrn1 binds RNA in concert with its active site and that mutation of the NLS domains alters transcription rates and half-lives of a subset of mRNA. However, the apparent overlap of the RNA binding domain and NLS complicates the separation of the RNA binding and shuttling activities and thus the interpretation of phenotypes ascribed to NLS domain mutants. Additionally, several experiments would benefit from additional controls to bolster their conclusions.

Thank you for viewing our experiments as “convincing”. The main concern of Reviewer 2 was “the apparent overlap of the RNA binding domain and NLS complicates the separation of the RNA binding and shuttling activities and thus the interpretation of phenotypes ascribed to NLS domain mutants.”

We agreed with this reservation and performed two experiments to resolve this issue, as indicated in our responses to the specific comments, below.

Major Points

1. The authors' main premise is that Xrn1 import is needed to mediate Kis mRNA regulation; however, from these data it is unclear whether Kis mRNA dysregulation in *xrn1ΔNLS1* cells is due to import defects or rather impaired binding of Xrn1ΔNLS1 to Kis mRNAs. The latter explanation is supported by Figure 3b and 3d, which suggest Tail and presumably NLS1 is a determinant of Xrn1 binding specificity. Therefore, the authors must find some strategy to establish the importance of Xrn1 shuttling without disrupting the NLS and/or demonstrate that NLS1 deletion does not differentially affect binding to different sets of RNA, especially Kis and non-Kis RNAs. For example, they could perform the RNA half-life analysis in a Kap120 mutant with wildtype Xrn1, and/or analyze Tail-GFP's binding profile to see whether Kis RNAs are enriched among the bound RNAs (especially RNAs with high RPKM values). Additionally, they should indicate whether the binding profile is highly correlated with total RNA abundance in yeast cells, which may suggest Tail binds RNA non-specifically.

We would like to thank Reviewer 2 for these suggestions, both of which we have adopted.

We first deleted *KAP120* from WT cells (expressing WT Xrn1) and observed a significant drop in both mRNA synthesis and decay (Fig. 4g and S3e and g; Table S3). Although mRNA synthesis and decay were affected, the average mRNA steady state

level was not changed (Fig. 4g and S3f), indicating that the effect on mRNA synthesis was balanced with that on mRNA decay. These results are similar to those observed in *xrn1*^{ΔNLS1/2} cells (cf Fig. 5a-b and S3e and g). Since Xrn1 in these *kap120Δ* cells was WT, and because these results are similar to those obtained with in *xrn1*^{ΔNLS1/2} strain, we conclude that the mutations that preventing Xrn1 import by two independent approaches affect mRNA synthesis and decay due to their effect on import, but not on RNA binding. Importantly, this conclusion agrees with our new results demonstrating that the mutations in NLS1/2 do not affect the overall binding of the purified full length Xrn1 with purified cellular mRNAs (Fig. 5j, “in vitro”), despite their importance for the RNA-binding capacity of the isolated tail. Probably the contribution of NLS1 to the overall RNA binding capacity of the full length Xrn1 is negligible and undetectable.

We have next analyzed our CRAC data to determine, as Reviewer 2 suggested, whether the binding profile of the isolated tail domain is, indeed, highly correlated with the total RNA abundance in yeast cells. The results are shown in Fig. S2e and discussed in lines 193-196. In addition, Venn diagram revealed that the isolated tail binds equally well Kis and non-Kis RNAs (Fig. S2f; and lines 193-196). These two results suggest that, when expressed outside Xrn1 context, Tail binds RNA non-specifically. Consistent with our conclusions, which are based on the two above mentioned experiments, these results (Fig. S2e-f) suggest that the mRNA decay bias (expressed as a “Kis value” – see lines 301-318) is not a trivial consequence of the RNA-binding feature of the tail domain.

2. The shuttling assays with the Nup49-313 mutant (Figure 2g) would be significantly bolstered by including a positive control for nuclear import using other shuttling proteins (e.g. Pab1) to show that the conditions used indeed cause an import defect. Also important is inclusion of a control to demonstrate that these conditions affect only import, since Figure 2h shows significantly greater Tail nuclear retention at the restrictive temperature, suggesting export may also be impacted.

We have included Rpb7-RFP, a shuttling Pol II subunit (Selitrennik et al., 2006 [ref 24]), as an internal positive control (Fig. 2g-h), which confirmed that the conditions used impair nuclear import. Thank you for this suggestion.

I do agree that our data suggest that inactivation of Nup49-313 affected the tail export. To the best of our knowledge, this is an unprecedented observation which warrants an investigation (a response to the heat shock? Import of export factors?), which is beyond the scope of this paper. In response to the minor comment #3, we now show the same experiment, except that instead of M457T we include the S454P (Fig. 2h). In this experiment, the nuclear localization after heat inactivation of Nup49-313 is like that at 24°C (the difference is not significant).

3. The data presented in Figure 4f do not strongly support the conclusion that RNA can outcompete Kap120 binding. The blot for Xrn1 seems overexposed, and the loading amount of Xrn1ΔNLS2 seems to decrease progressively, mirroring the change in Kap120. This should be addressed by decreasing the image exposure to confirm consistent loading and additionally denoting the statistical significance in the corresponding bar graph. The loading variation issue also applies to Figure S3C.

We also noticed that the amount of Xrn1^{ΔNLS2} decreases progressively. To determine whether this decrease mirrored the decrease in Xrn1, we have quantified the signals of both Kap120 and Xrn1 and expressed it as a ratio between the two proteins. This was done in 3 independent replicates and the statistics clearly support our conclusion. Similar quantification was done in Fig. S3c.

Minor Points

1. Further clarification/description of the logic leading to each set of experiments in the results section would help increase clarity.

We tried our best to respond to this suggestion. However, given the length of this manuscript, which has already exceeded the Journal allowance, we did it sparingly.

2. In general, the authors should clarify the number of cells counted for experiments (Fig1c-d) and include loading controls for Western blots from cell lysates.

This was done. Please see the figure legend.

3. Figure 2f should include TailM457TΔNLS1 mutant since M457T is the point mutant used for the subsequent shuttling experiments in Figure 2g. Alternatively, Figure 2g could include the S454P mutant.

We replaced M457T shown in Fig. 2g with the S454P mutant. We also changed panel h of this figure to show S454P. Thank you for noticing it.

4. Since graphs show discrete data points, they should be connected linearly, without smoothing.

This was done.

5. RNA binding to an NLS as a mechanism to regulate the shuttling of mRNA-associated proteins has been previously reported (e.g. PABPC shuttling in mammalian cells), and the discussion may benefit from drawing parallels to these examples.

We now discuss it in lines 518-525. Thank you.

Reviewer #3 (Remarks to the Author):

Here the authors report that sequence analyses of Xrn1 predict 2 NLS motifs, which are shown to be functional in GFP fusion constructs. NLS1 is located in an R3H-like motif, which is shown to contribute substantially to RNA-binding. This interesting finding suggests competition between karyopherin and RNA binding, as previously reported for ribosomal proteins (DOI: 10.1093/emboj/21.3.377). Consistent with the reduced RNA binding, mutation of the NLS stabilized a subset of mRNAs. Surprisingly, the authors attribute impaired cytoplasmic nuclease activity to loss of nuclear import, rather than impaired substrate binding, which seems the obvious potential explanation. The authors then interpret all subsequent analyses and the conclusion in terms of the model that nuclear import is a prerequisite for Xrn1 functions. This interpretation would be consistent with the data, but is not demonstrated to be correct.

The findings on karyopherin binding - and the potential competition with RNA binding and Xrn1 function are of interest and significance. I would be more enthusiastic about publication of a shorter paper that focused on these findings.

I would like to thank Reviewer 3 for carefully reading our manuscript, for finding our results interesting and significant and for the constructive comments and suggestions, in particular those related to the impact of NLS mutations on RNA binding.

Our responses to Reviewer 3's comments are indicated below (the reviewer's comments are in black characters, whereas our responses – in blue).

1: It was previously reported that “Rat1p and Xrn1p are functionally interchangeable exoribonucleases that are restricted to and required in the nucleus and cytoplasm, respectively” and that “targeting Xrn1p to the nucleus by the addition of the simian virus 40 large-T-antigen NLS resulted in complementation of the temperature sensitivity of a rat1-1 strain”. (Johnston Mol Cell Biol, 1997). The conclusion of the previous work was clearly that Xrn1 is functionally restricted to the cytoplasm. Were these findings simply incorrect? It would be useful for the authors to refute the earlier findings, since the conclusions are so directly opposed.

We do not think that Johnson results oppose ours. Arlen Johnson fused one of the strongest known heterologous NLS – that of the SV40 large T antigen - to Xrn1 and cloned it in a high-copy (2 μ) plasmid. This NLS is recognized by Kap95 importin β (see, for example Fig. 4c in [https://doi.org/10.1016/S0092-8674\(00\)80254-8](https://doi.org/10.1016/S0092-8674(00)80254-8)), which is different to the one that imports Xrn1. Using this construct, Johnson has found that expression of this plasmid-borne Xrn1 suppressed rat1-1 temperature sensitivity. This experiment suggested that nuclear 5' to 3' degradation activity is essential and that high level of nuclear Xrn1 (abnormally transported to the nucleus) can do the job. Normally, however, only a small portion of Xrn1 resides in the nucleus of dividing cells at any given time (see proportion of cells with nuclear Xrn1 in Fig. 1C). Xrn1 is localized in association with chromatin (Haimovich et al. 2013; Fischer et al. 2020). Equally important, this normal Xrn1 is imported to the nucleus by Kap120, whose Ran–GTP binding constant is the highest among the yeast karyopherins - 270 nM - as opposed to 0.23 nM of Kap95 (DOI: 10.1016/j.bbrc.2011.02.051). Since Ran–GTP binding releases

the cargo in the nucleus, it was proposed that, unlike Kap95 that releases its cargo as soon as it enters the nucleus, Kap120 does not release its cargo immediately after nuclear entry and the cargo is not readily available in the nucleus until its release (op. cit.). See also Fig. 7 and 8 and the illuminating discussion by Ed Hurt in doi:10.1093/emboj/17.8.2196., which deals with Mtr10-Npl3 complex that dissociates in the nucleus only when it arrives Pol II transcripts (Mtr10 Ran–GTP binding constant is similar to that of Kap120). Thus, normally, the small portion of the chromatin localized Xrn1 is not available to replace Rat1-1 under heat shock. A discussion about this comment is included in the revised manuscript (lines 457-466).

2: An obvious potential interpretation of the effects of mutations in Xrn1 on mRNA stability is that they block interactions with the substrate and/or other components of the cytoplasmic mRNA degradation machinery. If the authors want to conclude that this is not the case, they need to provide some direct support. In the model that nuclear pre-mRNAs are stoichiometrically loaded with Xrn1, a block on import should cause very rapid loss of 5' mRNA turnover, but not 3' degradation.

To address this important issue we have performed three new experiments as follows:

Experiment 1. To determine directly whether the mutations in the NLSs introduce any bias in binding capacity to intact mRNAs, we perform *in vitro* binding assay of Xrn1 and Xrn1 $\Delta^{\text{NLS1/2}}$ with purified total mRNAs followed by deep sequencing of the bound mRNAs – an *in vitro* RIP-seq. To evaluate the overall impact of Xrn1 import on both mRNA synthesis and decay, we assigned each gene with a “Kis value” that represents the sum of HL ratios ($\Delta^{\text{NLS1/2}}$ HL/WT HL) and inverse TR ratios (WT TR/ $\Delta^{\text{NLS1/2}}$ TR). This value better represents the effect of Xrn1 import on mRNA synthesis and decay than the binary distinction of “Kis” and “non-Kis” (see lines 301-319). Our results demonstrate that the two forms of Xrn1 bound mRNAs equally well; equal relative binding was observed for all Kis values (Fig. 5j, “*in vitro*”). This indicates that the mutations in the two NLSs do not detectably affect Xrn1 binding to mRNAs, despite the importance of NLS1 in the RNA-binding capacity of the isolated tail. Probably, the contribution of NLS1 to the overall RNA binding capacity of the full length Xrn1 is negligible and undetectable. In parallel, we performed a similar *in vivo* analysis, i.e., UV cross-linking of live cell followed by purifying Xrn1 or Xrn1 $\Delta^{\text{NLS1/2}}$, which revealed a Kis value sensitive differential interaction (Fig. 5j, “*in vitro*”). The difference between *in vitro* and *in vivo* results demonstrates that the interaction of Xrn1 with its substrates *in vivo* is not a trivial consequence of its interaction capacity with its substrate *in vitro*. This new experiment is described in lines 320-335.

Experiment 2. To examine the possibility that the mutations in NLS1/2 block interactions with other components of the cytoplasmic mRNA degradation machinery, we have affinity purified Xrn1-FLAG followed by mass spec of the co-purified proteins and compared it to those co-purified with Xrn1 $\Delta^{\text{NLS1/2}}$ -FLAG. As shown in Fig. S4m and Table S6, we found that Xrn1 $\Delta^{\text{NLS1/2}}$ binds equally well a number of mRNA decay factors (Dhh1 Dcp1, Dcp2 Edc3, Pat1, Ccr4 Not5). This experiment is discussed in lines 323-325.

Experiment 3. In response to Reviewer 2's suggestion, we deleted *KAP120* from WT cells (expressing WT Xrn1) and observed a significant drop in both mRNA synthesis and decay (Fig. 4g and S3e and g; Table S3). Although mRNA synthesis and decay were affected, the average mRNA steady state level was not changed (Fig. 4g and S3f), indicating that the effect on mRNA synthesis was balanced with that on mRNA decay. These results are similar to those observed in *xrn1*^{ΔNLS1/2} cells (cf Fig. 5a, b and i and Fig. S3e-g). Since Xrn1 in these *kap120*Δ cells was WT, and because these results are similar to those obtained with in *xrn1*^{ΔNLS1/2} strain, we conclude that the mutations that preventing Xrn1 import by two independent approaches affect mRNA synthesis and decay. This conclusion is further supported by our new results demonstrating that the mutations in NLS1/2 do not affect the overall binding of the purified full length Xrn1 with purified cellular mRNAs (Fig. 5j).

Taken together, mutations in Xrn1 NLSs do not affect Xrn1 capacity to bind other mRNA decay factors or its RNA substrates.

Please note our response to Reviewer 2 who suggested to “analyze Tail binding profile to see whether Kis RNAs are enriched among the bound RNAs (especially RNAs with high RPKM values)”, we used our RNA-binding data (shown in Fig. 3d) and found that the tail binds equally well Kis and non-Kis RNAs (see the Venn diagram in Fig. S2f), shown in the last page). Moreover, Reviewer 2 also suggested to “indicate whether the binding profile is highly correlated with total RNA abundance in yeast cells, which may suggest that, when expressed outside Xrn1 context, Tail binds RNA non-specifically”. We performed this analysis and found a significant correlation ($r=0.656$; $p<2.2e^{-16}$) (see Fig. S2e-f).

In the model that nuclear pre-mRNAs are stoichiometrically loaded with Xrn1, a block on import should cause very rapid loss of 5' mRNA turnover, but not 3' degradation.

We agree with this hypothesis. Our model, shown in Fig. 7, proposes that degradation of Kis mRNAs in *XRN1*^{ΔNLS1/2} cells is mediated by the exosome (Fig. 7, “nucleases other than Xrn1”). Strong support of this model is provided by the synthetic sickness of *SKI2* and *xrn1*^{ΔNLS1/2} (Fig. 5K). See lines 339-351.

3: The long, descriptive section on analyses of the cell cycle and growth conditions seems to be poorly connected to the first sections of the paper and adds little. In all of these analyses, the contributions of reduced substrate binding vs reduced karyopherin binding, and direct vs indirect effects of large-scale changes in mRNA metabolism, of would need to be clearly distinguished for conclusions to be drawn. This would be better followed up in a separate paper.

We would like to thank Reviewer 3 for this suggestion, which is tempting. After much deliberation, we decided to maintain Fig. 6, as it shows the biological relevance of the mRNA shuttling features that we reported in Figs 1-5.

4: It would be important to determine whether the proposed competition between important and RNA bunding actually takes place. Does karyopherin pre-binding inhibit RNA binding and degradation by Xrn1? Does RNA pre-binding block the karyopherin?

These would be significant additions to the paper.

We have performed the suggested experiment and found that Kap120 pre-binding did inhibit RNA binding. Please see Fig. S3d. This is added to our original Fig 4f, showing that RNA pre-binding blocks the karyopherin binding. The two experiments are described in lines 227-23. Thank you for this suggestion.

Minor point:

5: Page numbers would be helpful.

Numbers were included. Thank you noticing it.

REVIEWERS' COMMENTS

Reviewer #1 (Remarks to the Author):

Review for the manuscript NCOMMS-21-03251B by Dr. Choder and co-authors entitled "RNA-controlled nucleocytoplasmic shuttling of mRNA decay factors regulates mRNA synthesis and initiates a novel mRNA decay pathway".

The authors have addressed most of my previous concerns. I support the publication of the manuscript in Nature Communications.

Minor comments:

It might be better to clarify the nomenclature of Transcription rate (TR) in Fig 5a, Fig S3e, and synthesis rate (SR) in Fig4g.

Reviewer #2 (Remarks to the Author):

The authors have adequately addressed my prior issues through the inclusion of new data and text clarifications.

Reviewer #3 (Remarks to the Author):

The revised MS has been significantly improved and I am happy to recommend acceptance of this interesting report.